# Prompting ChatGPT in MNER: Enhanced Multimodal Named Entity Recognition with Auxiliary Refined Knowledge

**Jinyuan Li**[1], **Han Li**[2], **Zhuo Pan**[3], **Di Sun**[4], **Jiahao Wang**[3], **Wenkun Zhang**[5], **Gang Pan**[1,3,*]

[1]School of New Media and Communication, Tianjin University
[2]College of Mathematics, Taiyuan University of Technology
[3]College of Intelligence and Computing, Tianjin University
[4]Tianjin University of Science and Technology
[5]University of Copenhagen
{jinyuanli, wjhwtt, pangang}@tju.edu.cn, lihan0928@link.tyut.edu.com
panzhuotju@gmail.com, dsun@tust.edu.cn, jls704@alumni.ku.dk

## Abstract

Multimodal Named Entity Recognition (MNER) on social media aims to enhance textual entity prediction by incorporating image-based clues. Existing studies mainly focus on maximizing the utilization of pertinent image information or incorporating external knowledge from explicit knowledge bases. However, these methods either neglect the necessity of providing the model with external knowledge, or encounter issues of high redundancy in the retrieved knowledge. In this paper, we present PGIM — a two-stage framework that aims to leverage ChatGPT as an implicit knowledge base and enable it to heuristically generate auxiliary knowledge for more efficient entity prediction. Specifically, PGIM contains a Multimodal Similar Example Awareness module that selects suitable examples from a small number of predefined artificial samples. These examples are then integrated into a formatted prompt template tailored to the MNER and guide ChatGPT to generate auxiliary refined knowledge. Finally, the acquired knowledge is integrated with the original text and fed into a downstream model for further processing. Extensive experiments show that PGIM outperforms state-of-the-art methods on two classic MNER datasets and exhibits a stronger robustness and generalization capability.[1]

## 1 Introduction

Multimodal named entity recognition (MNER) has recently garnered significant attention (Lu et al., 2018). Users generate copious amounts of unstructured content primarily consisting of images and text on social media. The textual component in

---

[*]corresponding authors.
[1]Our code is publicly available at https://github.com/JinYuanLi0012/PGIM

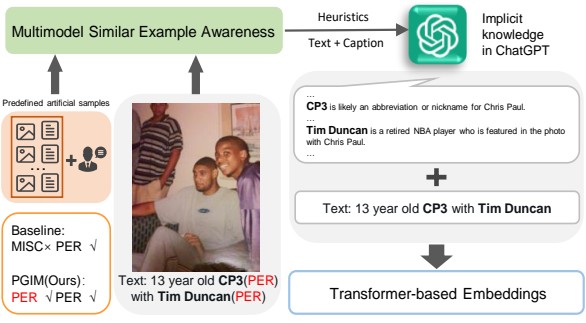

Figure 1: The "CP3" in the text is a class of entities that are difficult to predict successfully by existing studies. PGIM demonstrates successful prediction of such entities with an approach more similar to human cognitive processes by endowing ChatGPT with reasonable heuristics.

these posts possesses inherent characteristics associated with social media, including brevity and an informal style of writing. These unique characteristics pose challenges for traditional named entity recognition (NER) approaches (Chiu and Nichols, 2016; Devlin et al., 2018). To leverage the multimodal features and improve the NER performance, numerous previous works have attempted to align images and text implicitly using various attention mechanisms (Yu et al., 2020; Sun et al., 2021), but these Image-Text (I+T) paradigm methods have several significant limitations. Limitation 1. The feature distribution of different modalities exhibits variations, which hinders the model to learn aligned representations across diverse modalities. Limitation 2. The image feature extractors used in these methods are trained on datasets like ImageNet (Deng et al., 2009) and COCO (Lin et al., 2014), where the labels primarily consist of nouns rather than named entities. There are obvious deviations between the labels of these datasets and the named entities we aim to recognize. Given

these limitations, these multimodal fusion methods may not be as effective as state-of-the-art language models that solely focus on text.

While MNER is a multimodal task, the contributions of image and text modalities to this task are not equivalent. When the image cannot provide more interpretation information for the text, the image information can even be discarded and ignored. In addition, recent studies (Wang et al., 2021b; Zhang et al., 2022) has shown that introducing additional document-level context on the basis of the original text can significantly improve the performance of NER models. Therefore, recent studies (Wang et al., 2021a, 2022a) aim to solve the MNER task using the Text-Text (T+T) paradigm. In these approaches, images are reasonably converted into textual representations through techniques such as image caption and optical character recognition (OCR). Apparently, the inter-text attention mechanism is more likely to outperform the cross-modal attention mechanism. However, existing second paradigm methods still exhibit certain potential deficiencies:

(i) For the methods that solely rely on in-sample information, they often fall short in scenarios that demand additional external knowledge to enhance text comprehension.

(ii) For those existing methods that consider introducing external knowledge, the relevant knowledge retrieved from external explicit knowledge base (*e.g.*, Wikipedia) is too redundant. These low-relevance extended knowledge may even mislead the model's understanding of the text in some cases.

Recently, the field of large language models (LLMs) is rapidly advancing with intriguing new findings and developments (Brown et al., 2020; Touvron et al., 2023). On the one hand, recent research on LLMs (Qin et al., 2023; Wei et al., 2023; Wang et al., 2023a) shows that the effect of the generative model in the sequence labeling task has obvious shortcomings. On the other hand, LLMs achieves promising results in various NLP (Vilar et al., 2022; Moslem et al., 2023) and multimodal tasks (Yang et al., 2022; Shao et al., 2023). These LLMs with in-context learning capability can be perceived as a comprehensive representation of internet-based knowledge and can offer high-quality auxiliary knowledge typically. So we ask: *Is it possible to activate the potential of ChatGPT*

*in MNER task by endowing ChatGPT with reasonable heuristics?*

In this paper, we present **PGIM** — a conceptually simple framework that aims to boost the performance of model by Prompting ChatGPT In MNER to generate auxiliary refined knowledge. As shown in Figure 1, the additional auxiliary refined knowledge generated in this way overcomes the limitations of (i) and (ii). We begin by manually annotating a limited set of samples. Subsequently, PGIM utilizes the Multimodal Similar Example Awareness module to select relevant instances, and seamlessly integrates them into a meticulously crafted prompt template tailored for MNER task, thereby introducing pertinent knowledge. This approach effectively harnesses the in-context few-shot learning capability of ChatGPT. Finally, the auxiliary refined knowledge generated by heuristic approach of ChatGPT is subsequently combined with the original text and fed into a downstream text model for further processing.

PGIM outperforms all state-of-the-art models based on the Image-Text and Text-Text paradigms on two classical MNER datasets and exhibits a stronger robustness and generalization capability. Moreover, compared with some previous methods, PGIM is friendly to most researchers, its implementation requires only a single GPU and a reasonable number of ChatGPT invocations.

## 2 Related Work

### 2.1 Multimodal Named Entity Recognition

Considering the inherent characteristics of social media text, previous approaches (Moon et al., 2018; Zheng et al., 2020; Zhang et al., 2021; Zhou et al., 2022; Zhao et al., 2022) have endeavored to incorporate visual information into NER. They employ diverse cross-modal attention mechanisms to facilitate the interaction between text and images. Recently, Wang et al. (2021a) points out that the performance limitations of such methods are largely attributed to the disparities in distribution between different modalities. Despite Wang et al. (2022c) try to mitigate the aforementioned issues by using further refining cross-modal attention, training this end-to-end cross-modal Transformer architectures imposes significant demands on computational resources. Due to the aforementioned limitations, ITA (Wang et al., 2021a) and MoRe (Wang et al., 2022a) attempt to use a new paradigm to address MNER. ITA circumvents the challenge of multi-

modal alignment by forsaking the utilization of raw visual features and opting for OCR and image captioning techniques to convey image information. MoRe assists prediction by retrieving additional knowledge related to text and images from explicit knowledge base. However, none of these methods can adequately fulfill the requisite knowledge needed by the model to comprehend the text. The advancement of LLMs address the limitations identified in the aforementioned methods. While the direct prediction of named entities by LLMs in the full-shot case may not achieve comparable performance to task-specific models, we can utilize LLMs as an implicit knowledge base to heuristically generate further interpretations of text. This method is more aligned with the cognitive and reasoning processes of human.

## 2.2 In-context learning

With the development of LLMs, empirical studies have shown that these models (Brown et al., 2020) exhibit an interesting emerging behavior called In-Context Learning (ICL). Different from the paradigm of pre-training and then fine-tuning language models like BERT (Devlin et al., 2018), LLMs represented by GPT have introduced a novel in-context few-shot learning paradigm. This paradigm requires no parameter updates and can achieve excellent results with just a few examples from downstream tasks. Since the effect of ICL is strongly related to the choice of demonstration examples, recent studies have explored several effective example selection methods, *e.g.*, similarity-based retrieval method (Liu et al., 2021; Rubin et al., 2021), validation set scores based selection (Lee et al., 2021), gradient-based method (Wang et al., 2023b). These results indicate that reasonable example selection can improve the performance of LLMs.

## 3 Methodology

PGIM is mainly divided into two stages. In the stage of generating auxiliary refined knowledge, PGIM leverages a limited set of predefined artificial samples and employs the Multimodal Similar Example Awareness (MSEA) module to carefully select relevant instances. These chosen examples are then incorporated into properly formatted prompts, thereby enhancing the heuristic guidance provided to ChatGPT for acquiring refined knowledge. (detailed in §3.2). In the stage of entity prediction

based on auxiliary knowledge, PGIM combines the original text with the knowledge information generated by ChatGPT. This concatenated input is then fed into a transformer-based encoder to generate token representations. Finally, PGIM feeds the representations into the linear-chain Conditional Random Field (CRF) (Lafferty et al., 2001) layer to predict the probability distribution of the original text sequence (detailed in §3.3). An overview of the PGIM is depicted in Figure 2.

## 3.1 Preliminaries

Before presenting the PGIM, we first formulate the MNER task, and briefly introduce the in-context learning paradigm originally developed by GPT-3 (Brown et al., 2020) and its adaptation to MNER.

**Task Formulation** Consider treating the MNER task as a sequence labeling task. Given a sentence $T = \{t_1, \cdots, t_n\}$ with $n$ tokens and its corresponding image $I$, the goal of MNER is to locate and classify named entities mentioned in the sentence as a label sequence $y = \{y_1, \cdots, y_n\}$, where $y_i \in Y$ are predefined semantic categories with the BIO2 tagging schema (Sang and Veenstra, 1999).

**In-context learning in MNER** GPT-3 and its successor ChatGPT (hereinafter referred to collectively as GPT) are autoregressive language models pretrained on a tremendous dataset. During inference, in-context few-shot learning accomplishes new downstream tasks in the manner of text sequence generation tasks on frozen GPT models. Concretely, given a test input $x$, its target $y$ is predicted based on the formatted prompt $p(h, \mathcal{C}, x)$ as the condition, where $h$ refers to a prompt head describing the task and in-context $\mathcal{C} = \{c_1, \cdots, c_n\}$ contains $n$ in-context examples. All the $h, \mathcal{C}, x, y$ are text sequences, and target $y = \{y^1, \cdots, y^L\}$ is a text sequence with the length of $L$. At each decoding step $l$, we have:

$$y^l = \underset{y^l}{\arg\max}\, p_{\text{LLM}}(y^l | \boldsymbol{p}, y^{<l})$$

where LLM represents the weights of the pretrained large language model, which are frozen for new tasks. Each in-context example $c_i = (x_i, y_i)$ consists of an input-target pair of the task, and these examples is constructed manually or sampled from the training set.

Although the GPT-4[2] can accept the input of multimodal information, this function is only in the in-

---

[2] https://openai.com/product/gpt-4

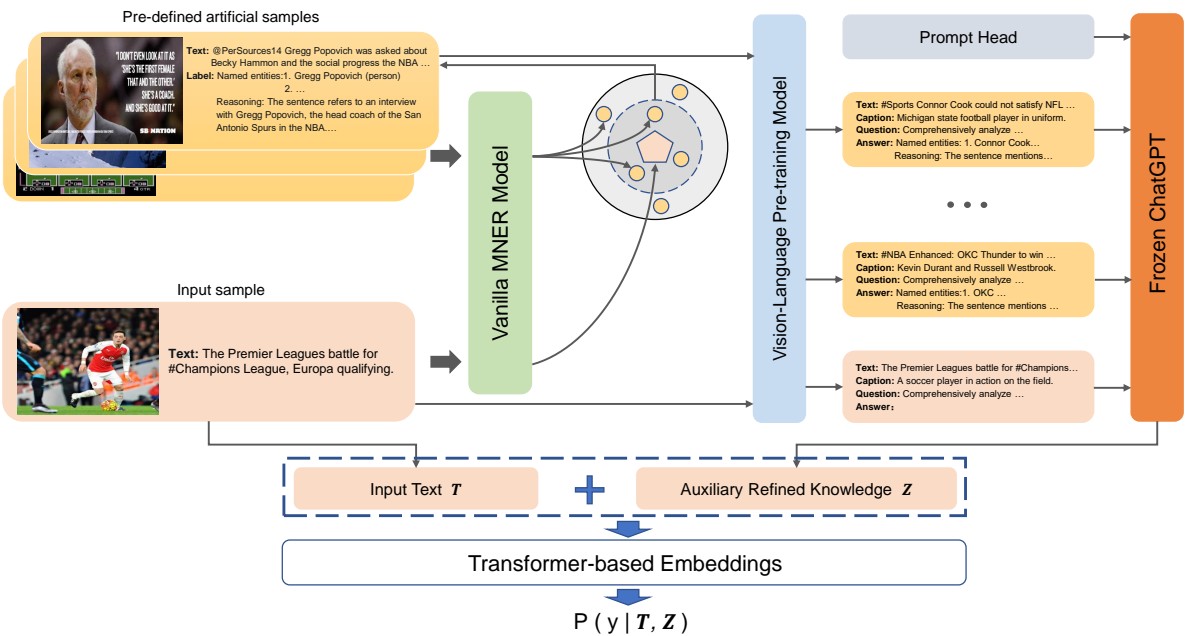

Figure 2: The architecture of PGIM.

ternal testing stage and has not yet been opened for public use. In addition, compared with ChatGPT, GPT-4 has higher costs and slower API request speeds. In order to enhance the reproducibility of PGIM, we still choose ChatGPT as the main research object of our method. And this paradigm provided by PGIM can also be used in GPT-4. In order to enable ChatGPT to complete the image-text multimodal task, we use advanced multimodal pre-training model to convert images into image captions. Inspired by PICa (Yang et al., 2022) and Prophet (Shao et al., 2023) in knowledge-based VQA, PGIM formulates the testing input $x$ as the following template:

```
Text: t \n Image: p \n Question: q \n Answer:
```

where $t$, $p$ and $q$ represent specific test inputs. \n stands for a carriage return in the template. Similarly, each in-context example $c_i$ is defined with similar templates as follows:

```
Text: t_i \n Image: p_i \n Question: q \n Answer: a_i
```

where $t_i$, $p_i$, $q$ and $a_i$ refer to an text-image-question-answer quadruple retrieved from predefined artificial samples. The complete prompt template of MNER consisting of a fixed prompt head, some in-context examples, and a test input is fed to ChatGPT for auxiliary knowledge generation.

## 3.2 Stage-1. Auxiliary Refined Knowledge Heuristic Generation

**Predefined artificial samples** The key to making ChatGPT performs better in MNER is to choose suitable in-context examples. Acquiring accurately annotated in-context examples that precisely reflect the annotation style of the dataset and provide a means to expand auxiliary knowledge poses a significant challenge. And directly acquiring such examples from the original dataset is not feasible.

To address this issue, we employ a random sampling approach to select a small subset of samples from the training set for manual annotation. Specifically, for Twitter-2017 dataset, we randomly sample 200 samples from training set for manual labeling, and for Twitter-2015 dataset, the number is 120. The annotation process comprises two main components. The first part involves identifying the named entities within the sentences, and the second part involves providing comprehensive justification by considering the image and text content, as well as relevant knowledge. For many possibilities encounter in the labeling process, what the annotator needs to do is to correctly judge and interpret the sample from the perspective of humans. For samples where image and text are related, we directly state which entities in the text are emphasized by the image. For samples where the image and text are unrelated, we directly declare that the image description is unrelated to the text. Through artifi-

cial annotation process, we emphasize the entities and their corresponding categories within the sentences. Furthermore, we incorporate relevant auxiliary knowledge to support these judgments. This meticulous annotation process serves as a guide for ChatGPT, enabling it to generate highly relevant and valuable responses.

**Multimodel Similar Example Awareness Module**   Since the few-shot learning ability of GPT largely depends on the selection of in-context examples (Liu et al., 2021; Yang et al., 2022), we design a Multimodel Similar Example Awareness (MSEA) module to select appropriate in-context examples. As a classic multimodal task, the prediction of MNER relies on the integration of both textual and visual information. Accordingly, PGIM leverages the fused features of text and image as the fundamental criterion for assessing similar examples. And this multimodal fusion feature can be obtained from various previous vanilla MNER models.

Denote the MNER dataset $\mathcal{D}$ and predefined artificial samples $\mathcal{G}$ as:

$$\mathcal{D} = \{(t_i, p_i, y_i)\}_{i=1}^{M}$$

$$\mathcal{G} = \{(t_j, p_j, y_j)\}_{j=1}^{N}$$

where $t_i, p_i, y_i$ refer to the text, image, and gold labels. The vanilla MNER model $\mathcal{M}$ trained on $\mathcal{D}$ mainly consists of a backbone encoder $\mathcal{M}_b$ and a CRF decoder $\mathcal{M}_c$. The input multimodal image-text pair is encoded by the encoder $\mathcal{M}_b$ to obtain multimodal fusion features $\mathcal{H}$:

$$\mathcal{H} = \mathcal{M}_b(t, p)$$

In previous studies, the fusion feature $\mathcal{H}$ after cross-attention projection into the high-dimensional latent space was directly input to the decoder layer for the prediction of the result. Unlike them, PGIM chooses $\mathcal{H}$ as the judgment basis for similar examples. Because examples approximated in high-dimensional latent space are more likely to have the same mapping method and entity type. PGIM calculates the cosine similarity of the fused feature $\mathcal{H}$ between the test input and each predefined artificial sample. And top-$N$ similar predefined artificial samples will be selected as in-context examples to enlighten ChatGPT generation auxiliary refined knowledge:

$$\mathcal{I} = \underset{j \in \{1,2,...,N\}}{\arg\text{TopN}} \frac{\mathcal{H}^T \mathcal{H}_j}{\|\mathcal{H}\|_2 \|\mathcal{H}_j\|_2}$$

$\mathcal{I}$ is the index set of top-$N$ similar samples in $\mathcal{G}$. The in-context examples $\mathcal{C}$ are defined as follows:

$$\mathcal{C} = \{(t_j, p_j, y_j) \mid j \in \mathcal{I}\}$$

In order to efficiently realize the awareness of similar examples, all the multimodal fusion features can be calculated and stored in advance.

**Heuristics-enhanced Prompt Generation**   After obtaining the in-context example $\mathcal{C}$, PGIM builds a complete heuristics-enhanced prompt to exploit the few-shot learning ability of ChatGPT in MNER.

A prompt head, a set of in-context examples, and a testing input together form a complete prompt. The prompt head describes the MNER task in natural language according to the requirements. Given that the input image and text may not always have a direct correlation, PGIM encourages ChatGPT to exercise its own discretion. The in-context examples are constructed from the results $\mathcal{C} = \{c_1, \cdots, c_n\}$ of the MSEA module. For testing input, the answer slot is left blank for ChatGPT to generate. The complete format of the prompt template is shown in Appendix A.4.

### 3.3   Stage-2. Entity Prediction based on Auxiliary Refined Knowledge

Define the auxiliary knowledge generated by ChatGPT after in-context learning as $Z = \{z_1, \cdots, z_m\}$, where $m$ is the length of $Z$. PGIM concatenates the original text $T = \{t_1, \cdots, t_n\}$ with the obtained auxiliary refining knowledge $Z$ as $[T; Z]$ and feeds it to the transformer-based encoder:

$$\{h_1, \cdots, h_n, \cdots, h_{n+m}\} = \text{embed}([T; Z])$$

Due to the attention mechanism employed by the transformer-based encoder, the token representation $H = \{h_1, \cdots, h_n\}$ obtained encompasses pertinent cues from the auxiliary knowledge $Z$. Similar to the previous studies, PGIM feeds $H$ to a standard linear-chain CRF layer, which defines the probability of the label sequence $y$ given the input sentence $T$:

$$P(y|T, Z) = \frac{\prod\limits_{i=1}^{n} \psi(y_{i-1}, y_i, h_i)}{\sum\limits_{y' \in Y} \prod\limits_{i=1}^{n} \psi(y'_{i-1}, y'_i, h_i)}$$

where $\psi(y_{i-1}, y_i, h_i)$ and $\psi(y'_{i-1}, y'_i, h_i)$ are potential functions. Finally, PGIM uses the negative

log-likelihood (NLL) as the loss function for the input sequence with gold labels $y^*$:

$$\mathcal{L}_{\text{NLL}}(\theta) = -\log P_\theta(y^*|T, Z)$$

## 4 Experiments

### 4.1 Settings

**Datasets** We conduct experiments on two public MNER datasets: Twitter-2015 (Zhang et al., 2018) and Twitter-2017 (Lu et al., 2018). These two classic MNER datasets contain 4000/1000/3257 and 3373/723/723 (train/development/test) image-text pairs posted by users on Twitter.

**Model Configuration** PGIM chooses the backbone of UMT (Yu et al., 2020) as the vanilla MNER model to extract multimodal fusion features. This backbone completes multimodal fusion without too much modification. BLIP-2 (Li et al., 2023) as an advanced multimodal pre-trained model, is used for conversion from image to image caption. The version of ChatGPT used in experiments is gpt-3.5-turbo and sampling temperature is set to 0. For a fair comparison, PGIM chooses to use the same text encoder XLM-RoBERTa$_{\text{large}}$ (Conneau et al., 2019) as ITA (Wang et al., 2021a), PromptM-NER (Wang et al., 2022b), CAT-MNER (Wang et al., 2022c) and MoRe (Wang et al., 2022a).

**Implementation Details** PGIM is trained by Pytorch on single NVIDIA RTX 3090 GPU. During training, we use AdamW (Loshchilov and Hutter, 2017) optimizer to minimize the loss function. We use grid search to find the learning rate for the embeddings within $[1 \times 10^{-6}, 5 \times 10^{-5}]$. Due to the different labeling styles of two datasets, the learning rates of Twitter-2015 and Twitter-2017 are finally set to $5 \times 10^{-6}$ and $7 \times 10^{-6}$. And we also use warmup linear scheduler to control the learning rate. The maximum length of the sentence input is set to 256, and the mini-batch size is set to 4. The model is trained for 25 epochs, and the model with the highest F1-score on the development set is selected to evaluate the performance on the test set. The number of in-context examples $N$ in PGIM is set to 5. All of the results are averaged from 3 runs with different random seeds.

### 4.2 Main Results

We compare PGIM with previous state-of-the-art approaches on MNER in Table 1. The first group of methods includes BiLSTM-CRF (Huang et al.,

2015), BERT-CRF (Devlin et al., 2018) as well as the span-based NER models (*e.g.*, BERT-span, RoBERTa-span (Yamada et al., 2020)), which only consider original text. The second group of methods includes several latest multimodal approaches for MNER task: UMT (Yu et al., 2020), UMGF (Zhang et al., 2021), MNER-QG (Jia et al., 2022), R-GCN (Zhao et al., 2022), ITA (Wang et al., 2021a), PromptMNER (Wang et al., 2022b), CAT-MNER (Wang et al., 2022c) and MoRe (Wang et al., 2022a), which consider both text and corresponding images.

The experimental results demonstrate the superiority of PGIM over previous methods. PGIM surpasses the previous state-of-the-art method MoRe (Wang et al., 2022a) in terms of performance. This suggests that compared with the auxiliary knowledge retrieved by MoRe (Wang et al., 2022a) from Wikipedia, our refined auxiliary knowledge offers more substantial support. Furthermore, PGIM exhibits a more significant improvement in Twitter-2017 compared with Twitter-2015. This can be attributed to the more complete and standardized labeling approach adopted in Twitter-2017, in contrast to Twitter-2015. Apparently, the quality of dataset annotation has a certain influence on the accuracy of MNER model. In cases where the dataset annotation deviates from the ground truth, accurate and refined auxiliary knowledge leads the model to prioritize predicting the truly correct entities, since the process of ChatGPT heuristically generating auxiliary knowledge is not affected by mislabeling. This phenomenon coincidentally highlights the robustness of PGIM. The ultimate objective of the MNER is to support downstream tasks effectively. Obviously, downstream tasks of MNER expect to receive MNER model outputs that are unaffected by irregular labeling in the training dataset. We further demonstrate this argument through a case study, detailed in the Appendix A.3.

### 4.3 Detailed Analysis

**Impact of different text encoders on performance** As shown in Table 2, We perform experiments by replacing the encoders of all XLM-RoBERTa$_{\text{large}}$ (Conneau et al., 2019) MNER methods with BERT$_{\text{base}}$ (Kenton and Toutanova, 2019). Baseline$_{\text{BERT}}$ represents inputting original samples into BERT-CRF. All of the results are averaged from 3 runs with different random seeds. The marker * refers to significant test p-value <

| Methods | Twitter-2015 | | | | | | | Twitter-2017 | | | | | | |
| | Single Type(F1) | | | | Overall | | | Single Type(F1) | | | | Overall | | |
| | PER | LOC | ORG | OTH. | Pre. | Rec. | F1 | PER | LOC | ORG | OTH. | Pre. | Rec. | F1 |
| --- | --- | --- | --- | --- | --- | --- | --- | --- | --- | --- | --- | --- | --- | --- |
| Text | | | | | | | | | | | | | | |
| BiLSTM-CRF[†] | 76.77 | 72.56 | 41.33 | 26.80 | 68.14 | 61.09 | 64.42 | 85.12 | 72.68 | 72.50 | 52.56 | 79.42 | 73.43 | 76.31 |
| BERT-CRF[‡] | 85.37 | 81.82 | 63.26 | 44.13 | 75.56 | 73.88 | 74.71 | 90.66 | 84.89 | 83.71 | 66.86 | 86.10 | 83.85 | 84.96 |
| BERT-SPAN[‡] | 85.35 | 81.88 | 62.06 | 43.23 | 75.52 | 73.83 | 74.76 | 90.84 | 85.55 | 81.99 | 69.77 | 85.68 | 84.60 | 85.14 |
| RoBERTa-SPAN[‡] | 87.20 | 83.58 | 66.33 | 50.66 | 77.48 | 77.43 | 77.45 | 94.27 | 86.23 | 87.22 | 74.94 | 88.71 | 89.44 | 89.06 |
| Text+Image | | | | | | | | | | | | | | |
| UMT | 85.24 | 81.58 | 63.03 | 39.45 | 71.67 | 75.23 | 73.41 | 91.56 | 84.73 | 82.24 | 70.10 | 85.28 | 85.34 | 85.31 |
| UMGF | 84.26 | 83.17 | 62.45 | 42.42 | 74.49 | 75.21 | 74.85 | 91.92 | 85.22 | 83.13 | 69.83 | 86.54 | 84.50 | 85.51 |
| MNER-QG | 85.68 | 81.42 | 63.62 | 41.53 | 77.76 | 72.31 | 74.94 | 93.17 | 86.02 | 84.64 | 71.83 | 88.57 | 85.96 | 87.25 |
| R-GCN | 86.36 | 82.08 | 60.78 | 41.56 | 73.95 | 76.18 | 75.00 | 92.86 | 86.10 | 84.05 | 72.38 | 86.72 | 87.53 | 87.11 |
| ITA | - | - | - | - | - | - | 78.03 | - | - | - | - | - | - | 89.75 |
| PromptMNER | - | - | - | - | 78.03 | 79.17 | 78.60 | - | - | - | - | 89.93 | 90.60 | 90.27 |
| CAT-MNER | 88.04 | **84.70** | 68.04 | 52.33 | 78.75 | 78.69 | 78.72 | 94.61 | 88.40 | 88.14 | **80.50** | 90.27 | 90.67 | 90.47 |
| MoRe_Text | - | - | - | - | - | - | 77.79 | - | - | - | - | - | - | 89.49 |
| MoRe_Image | - | - | - | - | - | - | 77.57 | - | - | - | - | - | - | 90.28 |
| MoRe_MoE | - | - | - | - | - | - | 79.21 | - | - | - | - | - | - | 90.67 |
| **PGIM(Ours)** | **88.34** | 84.22 | **70.15** | **52.34** | **79.21** | **79.45** | **79.33*** | **96.46** | **89.89** | **89.03** | 79.62 | **90.86** | **92.01** | **91.43*** |
| | ±0.02 | ±0.12 | ±0.36 | ±0.98 | ±0.63 | ±0.22 | ±0.06 | ±0.02 | ±0.68 | ±0.53 | ±2.25 | ±0.16 | ±0.07 | ±0.09 |

Table 1: Performance comparison on the Twitter-15 and Twitter-17 datasets. For the baseline model, results of methods with [†] come from Yu et al. (2020), and results with [‡] come from Wang et al. (2022c). The results of multimodal methods are all retrieved from the corresponding original paper. The marker * refers to significant test p-value < 0.05 when comparing with CAT-MNER and MoRe_Image/Text.

| | Twitter-2015 | | | Twitter-2017 | | |
| | Pre. | Rec. | F1 | Pre. | Rec. | F1 |
| --- | --- | --- | --- | --- | --- | --- |
| Baseline_BERT[◇] | 75.56 | 73.88 | 74.71 | 86.10 | 83.85 | 84.96 |
| UMT[†] | 71.67 | 75.23 | 73.41 | 85.28 | 85.34 | 85.31 |
| UMGF[†] | 74.49 | 75.21 | 74.85 | 86.54 | 84.50 | 85.51 |
| R-GCN[†] | 73.95 | 76.18 | 75.00 | 86.72 | 87.53 | 87.11 |
| ITA_BERT[†] | - | - | 75.60 | - | - | 85.72 |
| CAT_BERT[†] | **76.19** | 74.65 | 75.41 | 87.04 | 84.97 | 85.99 |
| MoRe_Image BERT[‡] | 73.16 | 74.64 | 73.89 | 85.49 | 86.38 | 85.94 |
| MoRe_Text BERT[‡] | 73.31 | 74.43 | 73.86 | 85.92 | 86.75 | 86.34 |
| PGIM_BERT | 75.84 | **77.76** | **76.79*** | 89.09 | 90.08 | 89.58* |
| | ±0.30 | ±0.22 | ±0.19 | ±0.24 | ±0.08 | ±0.10 |

Table 2: Results of methods with [†] are retrieved from the corresponding original paper. And results with [◇] come from Wang et al. (2022c).

0.05 when comparing with ITA, CAT-MNER and MoRe_Image/Text. And [‡] represents the results after we replace the text encoder in the MoRe official code with BERT_base.[3] The experimental results show that PGIM achieves a greater performance improvement than the XLM-RoBERTa_large version

experiment, especially on the Twitter-2017 dataset. We think the reasons for this phenomenon are as follows: XLM-RoBERTa_large conceals the defects of previous MNER methods through its strong encoding ability, and these defects are further amplified after using BERT_base. For example, the encoding ability of BERT_base on long text is weaker than XLM-RoBERTa_large, and the additional knowledge retrieved by MoRe_Image/Text is much longer than PGIM. Therefore, as shown in Table 2 and Table 5, the performance loss of MoRe_Image/Text is larger than the performance loss of PGIM after replacing BERT_base.

**Compared with direct prediction of ChatGPT**
Table 3 presents the performance comparison between ChatGPT and PGIM in the few-shot scenario. VanillaGPT stands for no prompting, and Prompt-GPT denotes the selection of top-N similar samples for in-context learning. As shown in Appendix A.4, we employ a similar prompt template and utilize the same predefined artificial samples as in-context examples for the MSEA module to select from. But the labels of these samples are replaced by named entities in the text instead of auxiliary knowledge.

[3] https://github.com/modelscope/AdaSeq/tree/master/examples/MoRe

| | Twitter-2015 | | | Twitter-2017 | | |
|---|---|---|---|---|---|---|
| | Pre. | Rec. | F1 | Pre. | Rec. | F1 |
| fs-10 | 0.16 | 12.00 | 0.32 | 0.26 | 12.95 | 0.51 |
| fs-50 | 50.09 | 54.50 | 52.20 | 49.40 | 51.96 | 50.65 |
| fs-100 | 57.33 | 66.26 | 61.47 | 69.51 | 74.09 | 71.73 |
| fs-200 | 65.16 | 73.57 | 69.11 | 80.97 | 85.05 | 82.96 |
| full-shot | 79.21 | 79.45 | 79.33 | 90.86 | 92.01 | 91.43 |
| VanillaGPT | 42.96 | 75.37 | 54.73 | 52.19 | 75.03 | 61.56 |
| PromptGPT$_{N=1}$ | 51.96 | 75.24 | 61.47 | 56.99 | 74.77 | 64.68 |
| PromptGPT$_{N=5}$ | 57.41 | 73.98 | 64.65 | 72.03 | 75.50 | 73.73 |
| PromptGPT$_{N=10}$ | 58.57 | 74.07 | 65.41 | 72.90 | 77.65 | 75.20 |

Table 3: Comparison of ChatGPT and PGIM in few-shot case. VanillaGPT and PromptGPT stand for direct prediction using ChatGPT.

| | Twitter-2015 | | | Twitter-2017 | | |
|---|---|---|---|---|---|---|
| | Pre. | Rec. | F1 | Pre. | Rec. | F1 |
| Baseline | 76.45 | 78.22 | 77.32 | 88.46 | 90.23 | 89.34 |
| w/o MSEA$_{N=1}$ | 78.15 | 79.01 | 78.58 | 90.49 | 90.82 | 90.65 |
| w/o MSEA$_{N=5}$ | 78.11 | **79.82** | 78.95 | 90.62 | 91.49 | 91.05 |
| w/o MSEA$_{N=10}$ | 78.47 | 79.21 | 78.84 | 90.54 | 91.77 | 91.15 |
| PGIM$_{N=1}$ | 78.40 | 79.21 | 78.76 | 89.90 | 91.63 | 90.76 |
| **PGIM$_{N=5}$** | **79.21** | 79.45 | **79.33** | **90.86** | 92.01 | **91.43** |
| PGIM$_{N=10}$ | 78.58 | 79.67 | 79.12 | 90.54 | **92.08** | 91.30 |

Table 4: Effect of the number of in-context examples on auxiliary refined knowledge.

The BIO annotation method is not considered in this experiment because it is a little difficult for ChatGPT. Only the complete match will be considered, and only if the entity boundary and entity type are both accurately predicted, we judge it as a correct prediction.

The results show that the performance of ChatGPT on MNER is far from satisfactory compared with PGIM in the full-shot case, which once again confirms the previous conclusion of ChatGPT on NER (Qin et al., 2023). In other words, when we have enough training samples, only relying on ChatGPT itself will not be able to achieve the desired effect. The capability of ChatGPT shines in scenarios where sample data are scarce. Due to the in-context learning ability of ChatGPT, it can achieve significant performance improvement after learning a small number of carefully selected samples, and its performance increases linearly with the increase of the number of in-context samples. We conduct experiments to evaluate the performance of PGIM in few-shot case. For each few-shot experiment, we randomly select 3 sets of training data and train 3 times on each set to obtain the average result. The results show that after 10 prompts, ChatGPT performs better than PGIM in the fs-100 scenario on both datasets. This suggests that ChatGPT exhibits superior performance when confronted with limited training samples.

**Effectiveness of MSEA Module**  Table 4 demonstrates the effectiveness of the MSEA module. We use the auxiliary refined knowledge generated by ChatGPT after $N$ in-context prompts to construct the datasets and train the model. The text encoder of the Baseline model is XLM-RoBERTa$_{large}$, and its input is the original text that does not contain any auxiliary knowledge. w/o MSEA represents a random choice of in-context examples. All results are averages of training results after three random initializations. Obviously, the addition of auxiliary refined knowledge can improve the effect of the model. And the addition of MSEA module can further improve the quality of the auxiliary knowledge generated by ChatGPT, which reflects the effectiveness of the MSEA module. An appropriate number of in-context examples can further improve the quality of auxiliary refined knowledge. But the number of examples is not the more the better. When ChatGPT is provided with an excessive number of examples, the quality of the auxiliary knowledge may deteriorate. One possible explanation for this phenomenon is that too many artificial examples introduce noise into the generation process of ChatGPT. As a pre-trained large language model, ChatGPT lacks genuine comprehension of the underlying logical implications in the examples. Consequently, an excessive number of examples may disrupt its original reasoning process.

**Case Study**  Through some case studies in Figure 3, we show how auxiliary refined knowledge can help improve the predictive performance of the model. The Baseline model represents that no auxiliary knowledge is introduced. MoRe$_{Text}$ and MoRe$_{Image}$ denote the relevant knowledge of the input text and image retrieved using text retriever and image retriever, respectively. In PGIM, the auxiliary refined knowledge generated by ChatGPT is structured into two components: the first component provides a preliminary estimation of the named entity, and the second component offers a corresponding contextual explanation. In these examples, "Leadership Course", "Big B", "Maxim"

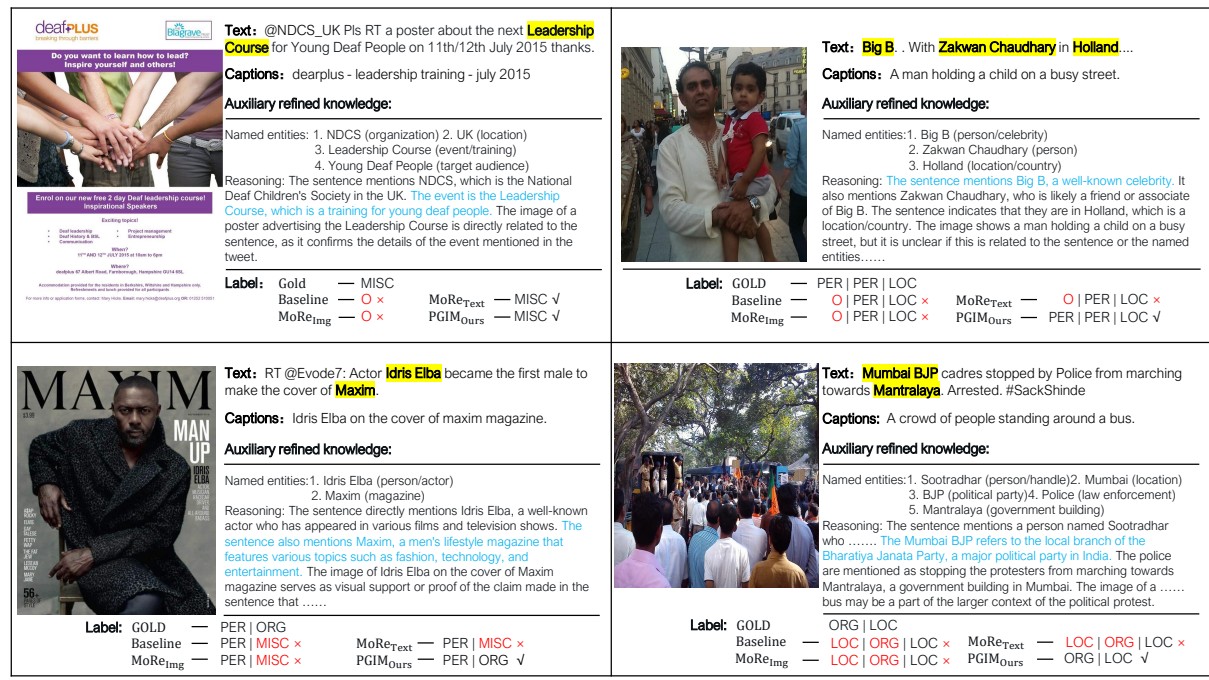

Figure 3: Four case studies of how auxiliary refined knowledge can help model predictions.

and "Mumbai BJP" are all entities that were not accurately predicted by past methods. Because our auxiliary refined knowledge provides explicit explanations for such entities, PGIM makes the correct prediction.

## 5 Conclusion

In this paper, we propose a two-stage framework called PGIM and bring the potential of LLMs to MNER in a novel way. Extensive experiments show that PGIM outperforms state-of-the-art methods and considerably overcomes obvious problems in previous studies. Additionally, PGIM exhibits a strong robustness and generalization capability, and only necessitates a single GPU and a reasonable number of ChatGPT invocations. In our opinion, this is an ingenious way of introducing LLMs into MNER. We hope that PGIM will serve as a solid baseline to inspire future research on MNER and ultimately solve this task better.

## Limitations

In this paper, PGIM enables the integration of multimodal tasks into large language model by converting images into image captions. While PGIM achieves impressive results, we consider this Text-Text paradigm as a transitional phase in the development of MNER, rather than the ultimate solution. Because image captions are inherently limited in their ability to fully capture all the details of an image. This issue may potentially be further resolved in conjunction with the advancement of multimodal capabilities in language and vision models (*e.g.*, GPT-4).

## Ethics Statement

In this paper, we use publicly available Twitter-2015, Twitter-2017 datasets for experiments. For the auxiliary refined knowledge, PGIM generates them using ChatGPT. Therefore, we trust that all data we use does not violate the privacy of any user.

## Acknowledgments

This work was supported by the Natural Science Foundation of Tianjin (No.21JCYBJC00640) and by the 2023 CCF-Baidu Songguo Foundation (Research on Scene Text Recognition Based on PaddlePaddle).

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

# A  Appendix

## A.1  Generalization Analysis

Due to the distinctive underlying logic of PGIM in incorporating auxiliary knowledge to enhance entity recognition, PGIM exhibits a stronger generalization capability that is not heavily reliant on specific datasets. Twitter-2015→2017 denotes the model is trained on Twitter-2015 and tested on Twitter-2017, vice versa. The results in Table 6 show that the generalization ability of PGIM is significantly improved compared with previous methods. This further validates the efficacy and superiority of our auxiliary refined knowledge in enhancing model performance.

## A.2  Comparison with MoRe

As the previous state-of-the-art method, MoRe retrieves relevant knowledge from Wikipedia to assist entity prediction. We experimentally compare the quality of auxiliary knowledge of MoRe and PGIM. The results are shown in Table 5. The baseline method solely relies on the original text without any incorporation of auxiliary information. $MoRe_{Text}$ and $MoRe_{Image}$ denote the relevant knowledge of the input text and image retrieved using text retriever and image retriever, respectively. Ave.length represents the average length of the auxiliary knowledge in entire dataset. Memory indicates the GPU memory size required for training model. Ave.Improve represents the average result

| Methods | Single Type(F1) | | | | Overall | | | Ave. length | Memory(MB) | Ave. Improve |
|---|---|---|---|---|---|---|---|---|---|---|
| | PER | LOC | ORG | OTH. | Pre. | Rec. | F1 | | | |
| | | | | | Twitter-2015 | | | | | |
| BaseLine | 87.04 | 83.49 | 67.34 | 50.16 | 76.45 | 78.22 | 77.32 | - | 11865 | - |
| MoRe$_{Text}$ | 86.92 | 83.08 | 68.20 | 49.15 | 77.12 | 77.77 | 77.45 | 227.41 | 16759 | ↓ 0.05 |
| MoRe$_{Image}$ | 87.38 | 83.78 | 67.75 | 49.38 | 77.44 | 78.06 | 77.75 | 203.00 | 16711 | ↑ 0.28 |
| **PGIM**(Ours) | **88.34** | **84.22** | **70.15** | **52.34** | **79.21** | **79.45** | **79.33** | **104.56** | **13901** | ↑ **1.86** |
| | | | | | Twitter-2017 | | | | | |
| BaseLine | 95.07 | 87.22 | 85.82 | 78.66 | 88.46 | 90.23 | 89.34 | - | 11801 | - |
| MoRe$_{Text}$ | 95.16 | 88.77 | 87.00 | 77.71 | 89.33 | 90.45 | 89.89 | 241.47 | 16695 | ↑ 0.50 |
| MoRe$_{Image}$ | 94.43 | 87.43 | 86.22 | 74.77 | 88.06 | 89.49 | 88.77 | 192.00 | 16447 | ↓ 0.80 |
| **PGIM**(Ours) | **96.46** | **89.89** | **89.03** | **79.62** | **90.86** | **92.01** | **91.43** | **94.52** | **13279** | ↑ **2.07** |

Table 5: Comparison of MoRe with PGIM. Since the original paper of MoRe (Wang et al., 2022a) did not report its Single Type (F1) on the Twitter-2015 and Twitter-2017 datasets, we run its released code and count the results. All of the results are averaged from 3 runs with different random seeds.

| | Twitter-2015→2017 | | | Twitter-2017→2015 | | |
|---|---|---|---|---|---|---|
| | Pre. | Rec. | F1 | Pre. | Rec. | F1 |
| UMT[†] | 67.80 | 55.23 | 60.87 | 64.67 | 63.59 | 64.13 |
| UMGF[†] | 69.88 | 56.92 | 62.74 | 67.00 | 62.81 | 66.21 |
| CAT-MNER[‡] | 70.69 | 59.44 | 64.58 | 74.86 | 63.01 | 68.43 |
| **PGIM** | **72.66** | **65.51** | **68.90** | **76.13** | **64.87** | **70.05** |

Table 6: Comparison of the generalization ability. For the baseline model, results with [†] come from Zhang et al. (2021), and results with [‡] come from Wang et al. (2022c).

after summing the improvement of each indicator compared with the baseline method. All models use XLM-RoBERTa$_{large}$ (Conneau et al., 2019) as the text backbone with a fixed batch size of 4. The experimental results demonstrate that PGIM achieves performance improvement while requiring shorter average auxiliary knowledge length and consuming less memory. This observation highlights the lightweight nature of PGIM and further underscores the superiority of our auxiliary refined knowledge compared with auxiliary knowledge of MoRe sourced from Wikipedia.

Additionally, we observe that in certain cases, the introduction of auxiliary knowledge by MoRe can even lead to a deterioration in model performance. One possible explanation for this phenomenon is that the information retrieved from Wikipedia often contains redundant or irrelevant content. The first case in Figure 4 illustrates this phenomenon well. In this case, PGIM makes the correct prediction because the information re-

trieved by ChatGPT clearly states that "Mumbai BJP refers to the Bharatiya Janata Party". However, the information retrieved by MoRe$_{Text}$ from Wikipedia provides almost no assistance in recognition of named entities. MoRe alleviates this problem to some extent by introducing the Mixture of Experts (MoE) module in the post-processing stage. They fixed the parameters of MoRe$_{Text}$ and MoRe$_{Image}$, and trained the MoE module for 50 epochs on the basis of them. But as shown in Table 1 before, compared with MoRe$_{MoE}$, PGIM still shows better results without any post-processing.

Furthermore, we also show an error prediction of PGIM in Figure 4. In this case, "Bush" is not a named entity that is hard to predict correctly. But since the additional knowledge retrieved by ChatGPT clearly states that "Bush 41" is a name of person, the prediction of PGIM is not in line with the gold label. This illustrates that the additional knowledge retrieved from ChatGPT can affect the final prediction of named entities to some extent. But the reason why MoRe$_{Text}$ can make correct prediction is obviously not related to the knowledge it retrieves from the Wikipedia, because "Bush" is not even mentioned in its knowledge. In fact, by using only the original text after masking the noise retrieved from the Wikipedia, the model can more easily predict correctly.

In summary, considering the relevance and length of retrieved information, using ChatGPT is obviously more suitable for this additional knowledge-based NER method than using Wikipedia. The information retrieved from Chat-

GPT is generally unambiguous and directional, which causes it to significantly help predictions in most cases, and may also mislead predictions in rare cases. But the information retrieved from Wikipedia may mislead the original predictions in many cases.

### A.3 Predictions for mislabeled examples

We observe that the annotation quality of the Twitter-2015 dataset is suboptimal. There have been a large number of errors and omissions in this dataset. This is the reason why the accuracy of Twitter-2015 has significantly decreased compared with Twitter-2017. However, as shown in Figure 5, since the first stage of ChatGPT heuristically generating auxiliary knowledge is not affected by mislabeling, PGIM correctly predicts those unlabeled entities. This also demonstrates the robustness of PGIM. As a future direction, we intend to reannotate the dataset to facilitate better development of the MNER task.

### A.4 Prompt template

We present the template for prompting ChatGPT to generate answers. In Figure 6, PGIM guides ChatGPT for auxiliary refined knowledge generation. In-context examples and answers in the template are selected from predefined artificial samples by the MSEA module. In Figure 7, we guide ChatGPT to make direct predictions. In-context examples are selected from the same predefined artificial samples by the MSEA module. Note that the answers here are no longer human answers, but named entities in text.

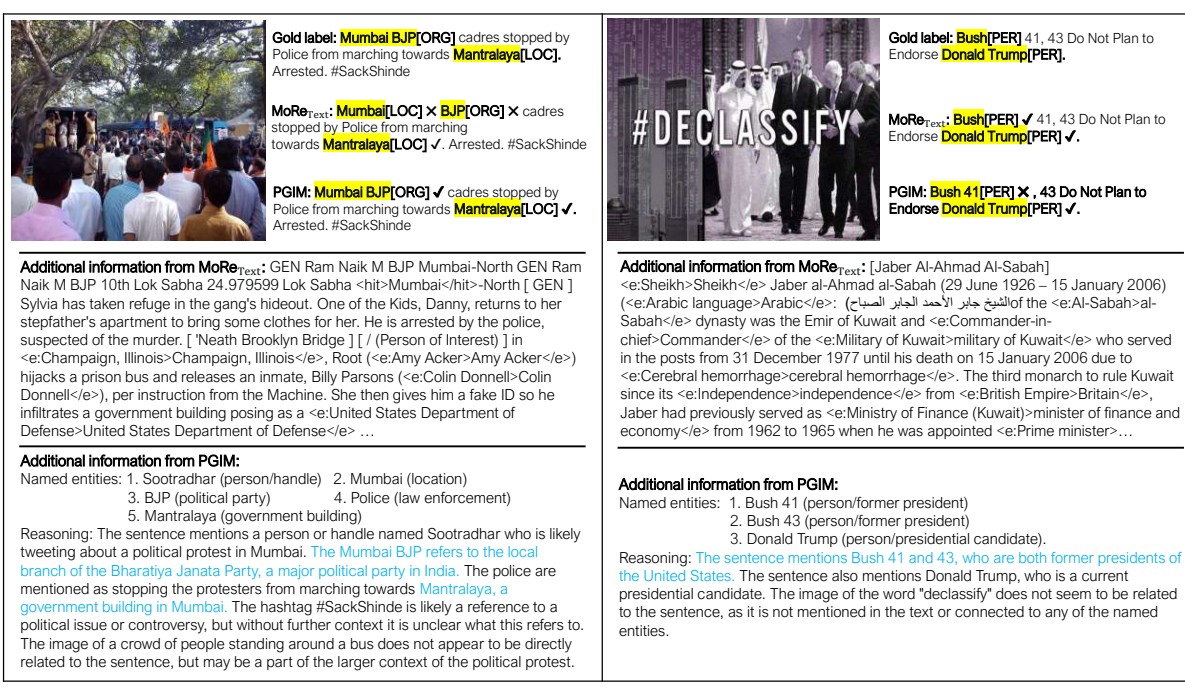

Figure 4: Two case studies on how information retrieved from Wikipedia by MoRe and information retrieved by PGIM from ChatGPT affects model predictions.

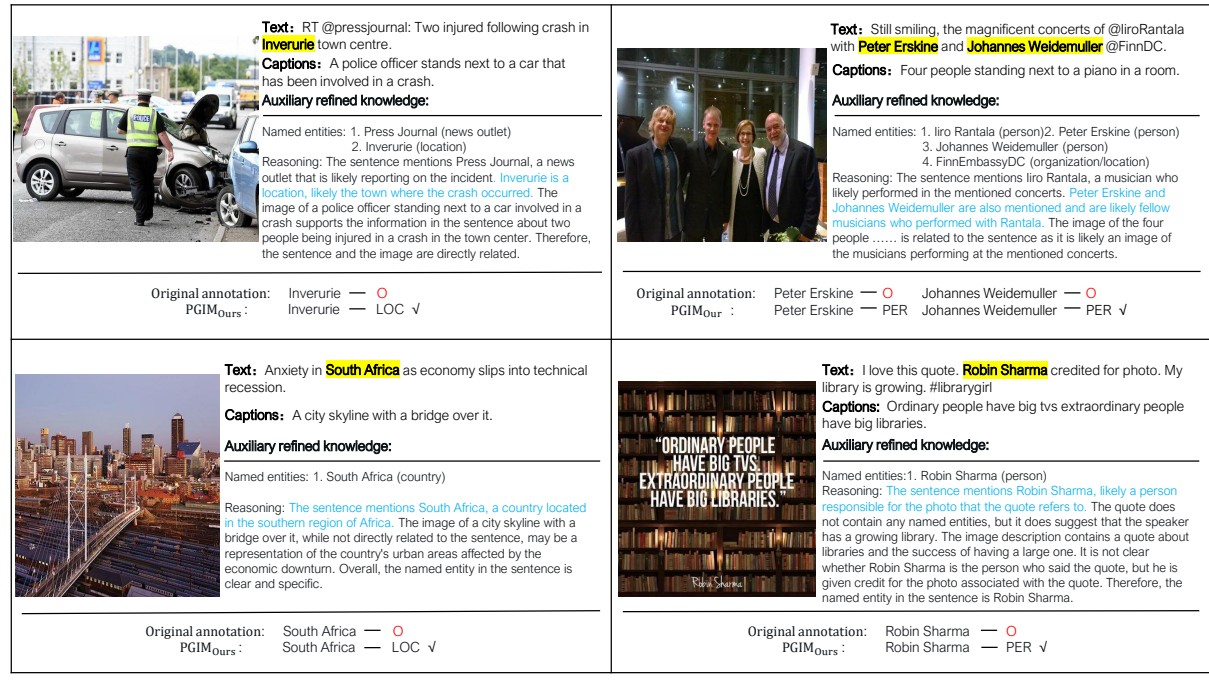

Figure 5: Some mislabeled examples of Twitter-2015 datasets.

## Prompt template for ChatGPT to make auxiliary explanation

Here are some content that people post on Twitter, and these content are composed of original text and image descriptions of the original text. Please note that the text and image descriptions here may or may not be relevant, so make your own judgment. Please follow the data annotation style and method reflected in the example I provided, comprehensively analyze the image description and the original text, determine which named entities and their corresponding types are included in the original text, and explain the reason for your judgment. Notice : just in 'Text', not include 'Image descriptions', don't change the writing style and format of entity names, and Words after the @ sign are not counted.

Text: #Sports Connor Cook could not satisfy NFL teams' questions about leadership.
Image descriptions: Michigan state football player in uniform.
Question: Comprehensively analyze the Text and the Image description, which named entities and their corresponding types are included in the Text? explain the reason for your judgment.
Answer: Named entities:1. Connor Cook (person/player) 2. NFL (league/organization) Reasoning: The sentence mentions Connor Cook, a former Michigan State football player who was drafted by the NFL. The NFL is the highest-level professional football league in the world. The image of a ......

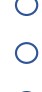

Text: #NBA Enhanced: OKC Thunder to win is NOW 10/3.
Image descriptions: Kevin Durant and Russell Westbrook.
Question: Comprehensively analyze the Text and the Image description, which named entities and their corresponding types are included in the Text? explain the reason for your judgment.
Answer: Named entities:1. OKC Thunder (team/franchise) 2. NBA (league/organization) Reasoning: The sentence mentions OKC Thunder, a professional basketball team based in Oklahoma City. The NBA is mentioned as the organization that the OKC Thunder are playing in. It is possible that the text and image are related in the sense that the odds for OKC Thunder to win may have been influenced by their past performances with the team.

Text: The Premier Leagues battle for #ChampionsLeague, Europa qualifying.
Image descriptions: A soccer player in action on the field.
Question: Comprehensively analyze the Text and the Image description, which named entities and their corresponding types are included in the Text? explain the reason for your judgment.
Answer:

Figure 6: A prompt template for ChatGPT to make auxiliary explanation.

## Prompt template for ChatGPT to direct predict

Here are some content that people post on Twitter, and these content are composed of original text and image descriptions of the original text. Please note that the text and image descriptions here may or may not be relevant, so make your own judgment. Please follow the data annotation style and method reflected in the example I provided, comprehensively analyze the image description and the original text, determine which named entities and their corresponding types are included in the original text. There will only be 4 types of entities: ['LOC', 'MISC', 'ORG', 'PER']. Make the answer format like: ['entity name1', 'entity type1'],['entity name2', 'entity type2']...... \n Notice : just in 'Text',not include 'Image descriptions', don't change the writting style and format of entity names in original Text, and the words beginning with @ sign are not counted.

Text: #Sports Connor Cook could not satisfy NFL teams' questions about leadership.
Image descriptions: Michigan state football player in uniform.
Question: Comprehensively analyze the Text and the Image description, which named entities and their corresponding types are included in the Text?
Answer: ['Connor Cook', 'PER'], ['NFL', 'ORG']

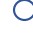

Text: #NBA Enhanced: OKC Thunder to win is NOW 10/3.
Image descriptions: Kevin Durant and Russell Westbrook.
Question: Comprehensively analyze the Text and the Image description, which named entities and their corresponding types are included in the Text?
Answer: ['OKC Thunder', 'ORG'], ['NBA', 'ORG']

Text: The Premier Leagues battle for #ChampionsLeague, Europa qualifying.
Image descriptions: A soccer player in action on the field.
Question: Comprehensively analyze the Text and the Image description, which named entities and their corresponding types are included in the Text?
Answer:

Figure 7: A prompt template for ChatGPT to direct predict.