# OpenReview forum: "Prompting ChatGPT in MNER: Enhanced Multimodal Named Entity Recognition with Auxiliary Refined Knowledge"
_EMNLP/2023/Conference — EMNLP 2023 Findings_

### Official Review · Reviewer_hWkn · 2023-08-01

**Soundness:** 3

**Excitement:**

3: Ambivalent: It has merits (e.g., it reports state-of-the-art results, the idea is nice), but there are key weaknesses (e.g., it describes incremental work), and it can significantly benefit from another round of revision. However, I won't object to accepting it if my co-reviewers champion it.

**Paper Topic And Main Contributions:**

This paper deals with the task of Multimodal Named Entity Recognition and proposes a two-stage framework PGIM that (1) aims to leverage ChatGPT as an implicit knowledge base and (2) utilize this generated auxiliary knowledge for MNER.
The first step utilizes manually annotated “artificial samples” from within a task’s training dataset and then for each data instance, identifies the K most similar artificial samples and uses those samples as in-context examples (text, image caption, gold labels) that get passed via a prompt template into ChatGPT along with the instance’s text and image, and a pre-defined prompt head (same for all instances). In the second step, the auxiliary refined knowledge output of the first stage, which is ChatGPTs few shot entities prediction for an input instance along with its reasoning, is fed into a transformer-based encoder along with the instance’s text and this output embedding is passed into standard linear-chain CRF layer for the final MNER prediction.  The transformer/CRF layer is fine tuned on the training data using negative log likelihood and then during inference time on test instances is used to make the final prediction.

The authors show that over two datasets ( Twitter 2015 and 2017 ) PGIM provides better performance than the current state of the art ( MoRe ) and run experiments/ablations to understand (1) the importance of how many artificial samples are annotated in total for stage 1, (2) how prompting GPT with top N similar samples ( without the subsequent transformer/CRF ) compares to PGIM and(3) the effect of random in-context examples vs the ones selected by PGIM (for different size n in [1,5,10]).  They finally share some qualitative examples to show how auxiliary refined knowledge can help model predictions (compared against a baseline model and MoRe )


**Questions For The Authors:**


Please see the prior section for my questions/concerns

**Reasons To Accept:**

1.  The authors present a simple and effective method to get state of the art results on the MNER task
2.  The ablations are thorough and justify many of the algorithmic decisions made.


**Reasons To Reject:**

(A) The paper doesn’t mention how many examples they needed to manual annotate per training dataset ( but we know that its greater than 200 .. so possibly 300, 500, 1k, 4k? ).
      This is one of the most important parts of the paper and needs to be provided so people can understand the tradeoff ( simplicity vs annotation effort).
      The method is well justified and simple, but the way in which they annotate and how many instances they need to annotate is opaque, in particular the second part which “involves providing comprehensive justification by considering the image and text content, as well as relevant knowledge.”.  Are there any examples of this annotation that you can share?  Will this data be provided?

(B) Their results against MoRe are fairly close ( .5 and .3 ) and the appendix comparison ( which is not mentioned in the body of the paper ) doesn’t compare against the full MoRe but rather just how well it does if it only uses
one modality ( text or image ).  I think there is a probably a good case to make ( efficiency, cost, controllability, etc vs annotation costs ) for why to use PGIM over MoRe, but they should articulate it in the paper and compare more truly along each entity type if possible ( and not just the final F1 scores )

(C) It would be nice to see how this method would do using an open source alternative ( alpaca-lora, vicuna-13B, llama2, etc) and not just ChatGPT for comparison.

(D) Its great that all experiments are averaged over 3 runs, but including error bars would be useful to determine the variance of things ( even if only in the appendix for space reasons )

(E) For the case study, it would be interesting to do an error analysis against MoRe results in particular ( when is it right and PGIM is wrong and vice versa, when is having Wiki access advantageous vs ChatGPT , etc)



**Reproducibility:**

5: Could easily reproduce the results.

**Reviewer Confidence:**

4: Quite sure. I tried to check the important points carefully. It's unlikely, though conceivable, that I missed something that should affect my ratings.

**Typos Grammar Style And Presentation Improvements:**


- Line 214, the first time you mention CRF you should spell out Conditional Random Field and cite it.
- The end of the Task formulation sentence “y \elem Y with Y which predefined .. “ (228-231) is a little weirdly worded.  Maybe “y \elem Y are predefined … “
- Line 819 small typo (Ave Imporve -> Ave Improve)

---

> ### Author Rebuttal · Authors · 2023-08-28
>
> We extend our gratitude for all the highly constructive feedback you've provided, and we particularly appreciate your thorough review of our paper. We will address your concerns as follows:
>
> **Q(A): The paper doesn’t mention how many examples they needed to manual annotate per training dataset. This is one of the most important parts of the paper and needs to be provided so people can understand the tradeoff. The method is well justified and simple, but…**
>
> A: First, about predefined artificial samples: specifically, for Twitter-2017 dataset, we randomly sample 200 samples from the training set for manual labeling, and for Twitter-2015 dataset, the number is 120. The annotators consist of three NLP graduate students with backgrounds in NER and LLMs. For many possibilities encountered in the labeling process, what the annotator needs to do is to correctly judge and interpret the sample from the perspective of humans. For samples where images and text are related, we directly state which entities in the text are emphasized by the image. For samples where images and text are not related, we directly point out that the two are not related.
>
> Second, although the MNER work in the past two years has rarely open-sourced data and codes, after the anonymity period is over, we will open source all the data of PGIM (codes, datasets, predefined artificial samples) to GitHub. Currently, you can view all the data and code (including predefined artificial samples) related to PGIM through Supplementary Materials.
>
> Your valuable comments made us realize that a more detailed explanation of the notation process of predefined artificial samples can better help readers understand paper. So, the above replies will eventually be condensed in the camera-ready version and added to paragraph 3.2 Predefined artificial samples.
>
> **Q(B): Their results against MoRe are fairly close ( .5 and .3 ) and the appendix comparison ( which is not mentioned in the body of the paper ) doesn’t compare against the full MoRe but rather just how well it does if it only uses one modality ( text or image ). I think there is a probably a good case to make ( efficiency, cost, controllability, etc vs annotation costs ) for why to use PGIM over MoRe, but they should articulate it in the paper and compare more truly along each entity type if possible ( and not just the final F1 scores )**
>
> A: First, we need to point out that the fair opponent of PGIM should be MoRe (Image/Text) in the Appendix Table 4. Because they have the same macrostructure, same parameter quantity and same encoder (RoBERTa-large). The only difference is the method of obtaining additional information and the quality of the acquired knowledge. What Table 1 really wants to reflect is: as we stated in lines 833-846 of the paper, even when making an unfair comparison with MoRe (MoE), which is built upon MoRe (Image+Text) by incorporating post-processing steps-MoE (This means that MoRe[4] needs to train three models: Image+Text+MoE). PGIM still achieves better results without any excessive complex design (Just need to train one model).
>
> Therefore, compared with the complete MoRe (MoE), the primary reason for choosing PGIM is that even though the performance of MoRe (MoE) is competitive, the training difficulty of PGIM and the requirements for hardware are much lower than that of MoRe (MoE). In fact, the parameter quantity of PGIM is the same as that of MoRe(Image) or MoRe(Text). This means that the model parameters of the complete MoRe (MoE), are nearly three times that of PGIM (Image module + Text module + MoE module). Obviously, the training difficulty of MoRe (MoE) and the requirements for hardware far exceed that of PGIM. In addition, MoRe[4] did not open source the data and code of its MoE part. And in the past two years, there has been a lack of open source research in the field of MNER. We hope that PGIM will be easy to use and become a solid baseline for the future of the field of MNER.
>
> Second, when making a fair comparison (same architecture and model parameters) with publicly available MoRe (Image/Text), the key reasons for choosing PGIM are as follows: as shown in Table 4, compared with MoRe (Image/Text), PGIM achieves a more significant performance improvement (about 2%) with a shorter appended text length on all entity types. In addition, it is worth mentioning that the shorter additional text length in PGIM naturally results in a smaller memory requirement. Less than 16GB training memory requirement (in condition without changing float precision and batchsize) means that PGIM can be trained and fine-tuned on a wider range of GPUs than MoRe (Image/Text) (e.g., V100 16GB/4080 16GB). This lightweight advantage of PGIM is not present in all MoRe (Image/Text/MoE).
>
> But, for PGIM and MoRE (Image/Text) with similar architectures, what about using a more lightweight text encoder? As requested by the reviewer KB7s, we also conducted an experiment. Text encoders for all methods in Table 6 are BERT-base. Baseline$_{BERT}$ represents inputting original samples into BERT+CRF. All of the results are averaged from 3 runs with different random seeds. The marker * refers to significant test p-value < 0.05 when comparing with CAT-MNER and MoRe(Image/Text). When we switch the text encoder from RoBERTa-large to BERT-base, PGIM gains a much larger lead (about 3%~4%) over MoRe(Image/Text).  This is because BERT's long-text encoding capability is significantly inferior to that of RoBERTa. Consequently, the disadvantage of MoRe's excessively long additional text is further magnified in this context.
>
> |           Table 6     |  |**Twitter-2015**     |     | |    **Twitter-2017**  |     |
> |-----------------------|:----------------:|:---:|:---:|:----------------:|:---:|:---:|
> |                       |       Pre.       | Rec.|  F1  |       Pre.       | Rec.|  F1  |
> | Baseline$_{BERT}$ |      75.56      |73.88|74.71 |      86.10       |83.85|84.96 |
> | UMT                   |      71.67      |75.23|73.41 |      85.28       |85.34|85.31 |
> | UMGF                  |      74.49      |75.21|74.85 |      86.54       |84.50|85.51 |
> | R-GCN                 |      73.95      |76.18|75.00 |      86.72       |87.53|87.11 |
> | ITA$_{BERT}$   |        -        |  -  |75.60 |        -         |  -  |85.72 |
> | CAT-MNER$_{BERT}$ |    **76.19**    |74.65|75.41 |      87.04       |84.97|85.99 |
> | MoRe$_{Image BERT}$ |  73.16  |74.64|73.89 |      85.49       |86.38|85.94 |
> | MoRe$_{Text BERT}$ |  73.31  |74.43|73.86 |      85.92       |86.75|86.34 |
> | PGIM$_{BERT}$  |      75.84      |**77.76**|**76.79*** |      **89.09**      |**90.08**|**89.58*** |
> |                       | $\pm$0.30 |$\pm$0.22|$\pm$0.19 | $\pm$0.24 |$\pm$0.08|$\pm$0.10|
>
> Third, the results of Table 4 and Table 6 also show an interesting result: As described in lines 833-846 of the paper, the performance of MoRe (Image/Text) in some cases is even worse than that of Baseline without any auxiliary knowledge. In other words, using MoRe (Image/Text) to retrieve the additional knowledge from the wiki is not even as good as using the original text alone. The information retrieved by MoRe from wiki is messy, invalid and redundant, which leads to the fact that the original text features finally fed to the decoder contain too much irrelevant noise. This is also one of the core arguments of PGIM, as effective as possible and short text additional knowledge is really effective for NER.
>
> In addition, according to your valuable suggestions, we also made statistics on the prediction results of PGIM and MoRe (Text) on all entity categories, the results are as follows:
>
> | Tabel 7     | Error$_{MoRe}$ | Right$_{PGIM}$ | Accurate | Error$_{PGIM}$ | Right$_{MoRe}$ | Accurate |
> |-------|-----------------------|-----------------------|----------|-----------------------|-----------------------|----------|
> |  |                       |                       |   **Twitter-2017**       |                       |                       |          |
> | PER   | 69                    | 36                    | 52.17%   | 59                    | 15                    | 25.42%   |
> | LOC   | 30                    | 14                    | 46.66%   | 23                    | 6                     | 26.08%   |
> | ORG   | 116                   | 54                    | 46.55%   | 84                    | 25                    | 29.76%   |
> | OTHER | 87                    | 38                    | 43.68%   | 60                    | 4                     | 0.07%    |
> | |                       |                       |      **Twitter-2015**     |                       |                       |          |
> | PER   | 335                   | 129                   | 38.51%   | 246                   | 40                    | 16.26%   |
> | LOC   | 320                   | 87                    | 27.19%   | 316                   | 71                    | 22.47%   |
> | ORG   | 426                   | 99                    | 23.24%   | 417                   | 84                    | 20.14%   |
> | OTHER | 819                   | 169                   | 20.64%   | 738                   | 88                    | 11.92%   |
>
> Since the performance of MoRe(Text) and MoRe(Image) are similar, considering the space reason, only the comparison results between PGIM and MoRe(Text) are listed here. **Error(X)** represents the number of wrong predictions made by the **X** model, **Right(Y)** represents the number of correct predictions made by the **Y** model among all samples where the **X** model made wrong predictions, and **Accurate=Right(Y)/Error(X)**. The results show that on the 2017 dataset, PGIM is nearly twice as accurate as MoRe when facing all error-prone samples of PER, LOC, and ORG. In the face of the most difficult OTHER category, the accuracy rate has achieved a crushing advantage. In other words, in the face of PGIM's error-prone samples, MoRe basically cannot predict correctly, and in the face of MoRe's error-prone samples, PGIM can make correct predictions in half of the cases. On the 2015 dataset with relatively irregular labels, PGIM still won the overall lead in four categories. Although the lead has diminished somewhat, the accuracy of PGIM still remains twice that of MoRe when facing the PER and OTHER categories.
>
> Finally, we need to point out: the annotation cost of PGIM is extremely low. In Table 3, w/o MSEA$_{N=1,5,10}$ shows the performance of PGIM on randomly selected N artificial samples, and its performance is at the same level as that of the full MoRe (Image+Text+MoE). In other words, when faced with a new dataset, it only needs to simply and quickly construct a few artificial samples adapted to the new dataset to obtain the strong performance shown by PGIM (w/o MSEA). The reason for more artificial samples is to explore the possibility of obtaining marginal gains in performance through the MSEA module + artificial samples. Annotation costs are negligible when there is no need to pursue extreme performance.
>
> Your suggestion made us realize that it is very important to further explain the advantages of PGIM over MoRe(Image/Text/MoE) in the paper. All of the above analyzes will eventually be condensed and added to the Appendix of the paper in camera-ready version.
>
> **Q(C): It would be nice to see how this method would do using an open source alternative (alpaca-lora, vicuna-13B, llama2, etc) and not just ChatGPT for comparison.**
>
> A: Thank you for your valuable comments. We have tried to use multimodal LLMs such as MiniGPT-4[8]/mPLUG-Owl[9], and we finally chose ChatGPT mainly because of the following two points:
>
> First, we found that the most obvious defect of this kind of open source alternative compared to ChatGPT is that the operability of its prompt project is much lower than that of ChatGPT/GPT4. The effect of a randomly selected prompt + example is not much worse than that of a well-designed prompt + example. This is contrary to the starting point of PGIM, which aims to explore a prompting approach tailored for MNER.
>
> Second, when it comes to comparing PGIM with LLMs for direct prediction of named entities. For open source alternatives that can be deployed locally, it is difficult for us to control its persistence to generate answers in a specified format. In the actual test, about 5% of chatgpt's answers did not follow the format in the prompt, and this value was even close to 30% on mPLUG-Owl[9] and MiniGPT-4[8]. Irregular answers will make it impossible to use the program to count the correct rate of the results. Obviously, ChatGPT is the most powerful LLM, and PGIM is far better than ChatGPT, which is enough to achieve the goal of the experiment in Table 2.
>
> Your opinion has inspired us very well. Due to time constraints, we cannot complete the evaluation of various open source LLMs within five days, because it requires a huge workload. We believe that the exhaustive experiments in this paper have provided sufficient evidence for our claim. But we will do our best to test before camera ready. Perhaps in MNER task, how to effectively prompt on LLMs that can be deployed locally such as llama2, is another question that is worthy of our exploration in the future.
>
> **Q(D): Its great that all experiments are averaged over 3 runs, but including error bars would be useful to determine the variance of things (even if only in the appendix for space reasons)**
>
> A: You've presented excellent insights. According to statistics, the standard deviation of PGIM in Table 1 are as follows:
>
> | Tabel 1${_{add SD}}$ | ******|Twitter|-   2015 |******|******|******|****** | ******|Twitter|- 2017  |****** |****** |****** | ******|
> | --- | --- | --- | --- | --- | --- | --- | --- | --- | --- | --- | --- | --- | --- | --- |
> |  | | Single|  Type(F1)| | |Overall | | |Single |Type(F1) | | Overall | | |
> | | PER | LOC  | ORG | OTH. | Pre. | Rec. | F1 | PER | LOC | ORG | OTH. | Pre. | Rec. | F1 |
> | | | | | | | |Text+Image | | | | | | | |
> | UMT | 85.24    | 81.58 | 63.03 | 39.45 | 71.67 | 75.23 | 73.41 | 91.56 | 84.73 | 82.24 | 70.10 | 85.28 | 85.34 | 85.31 |
> | UMGF | 84.26 | 83.17 | 62.45 | 42.42 | 74.49 | 75.21 | 74.85 | 91.92 | 85.22 | 83.13 | 69.83 | 86.54 | 84.50 | 85.51 |
> | MNER-QG | 85.68 | 81.42 | 63.62 | 41.53 | 77.76 | 72.31 | 74.94 | 93.17 | 86.02 | 84.64 | 71.83 | 88.57 | 85.96 | 87.25 |
> | R-GCN | 86.36 | 82.08 | 60.78 | 41.56 | 73.95 | 76.18 | 75.00 | 92.86 | 86.10 | 84.05 | 72.38 | 86.72 | 87.53 | 87.11 |
> | ITA | - | - | - | - | - | - | 78.03 | - | - | - | - | - | - | 89.75 |
> | PromptMNER | - | - | - | - | 78.03 | 79.17 | 78.60 | - | - | - | - | 89.93 | 90.60 | 90.27 |
> | CAT-MNER | 88.04 | **84.70** | 68.04 | 52.33 | 78.75 | 78.69 | 78.72 | 94.61 | 88.40 | 88.14 | **80.50** | 90.27 | 90.67 | 90.47 |
> | MoRe${_{MoE}}$ | - | - | - | - | - | - | 79.21 | - | - | - | - | - | - | 90.67 |
> | PGIM(Ours) | **88.34** | 84.22 | **70.15** | **52.34** | **79.21** | **79.45** | **79.33** | **96.46** | **89.89** | **89.03** | 79.62 | **90.86** | **92.01** | **91.43** |
> | | $\pm$0.02 | $\pm$0.12 | $\pm$0.36 | $\pm$0.98 | $\pm$0.63 | $\pm$0.22 | $\pm$0.06 | $\pm$0.02 | $\pm$0.68 | $\pm$0.53 | $\pm$2.25 | $\pm$0.16 | $\pm$0.07 | $\pm$0.09 |
>
> In fact, we have indeed attempted to display the standard deviation in the Table1, as it holds significant importance. But considering that most of the methods in Table 1 (UMT[5], UMGF[6], MNER-QG[7], ITA[1], PromptMNER[2], CAT-MNER[3], MoRe[4]) did not report the standard deviation in their paper, we ended up following this "convention" in MNER.
>
> In addition, it is very worth mentioning that ITA[1] pointed out that since the baseline method actually selects the test set for parameter adjustment and reports the best result instead of the average result, therefore the final report of ITA[1] is the best accuracy rather than the average accuracy of multiple experiments. Accordingly, this may be the reason why the research in the field of MNER recently (UMT[5], UMGF[6], MNER-QG[7], ITA[1], PromptMNER[2], CAT-MNER[3], MoRe[4]) does not report the standard deviation.
>
> This problem exists in almost all the work in MNER field. But we hope that PGIM can be the terminator of this problem, as a solid baseline in the field of MNER. So we will modify the results in the camera ready version to clearly present the standard deviation.
>
> **Q(E): For the case study, it would be interesting to do an error analysis against MoRe results in particular (when is it right and PGIM is wrong and vice versa, when is having Wiki access advantageous vs ChatGPT, etc)**
>
> A: We analyzed the advantages of PGIM from the level of statistical indicators in **Q(B)**. In case the statistics are not intuitive enough, we will illustrate through a more intuitive case analysis.
>
> The first is a situation where MoRe(Text) makes a mistake but PGIM gets it right when faced with the complex sample. For the fourth most complex sample in Figure 4 of paper, here is the additional information MoRe(Text) retrieved from the wiki and the predictions it made:
> _______________________________________________________________________________________________
> **Additional information from wiki:** GEN Ram Naik M BJP Mumbai-North GEN Ram Naik M BJP 10th Lok Sabha 24.979599 Lok Sabha <hit>Mumbai</hit>-North [ GEN ] Sylvia has taken refuge in the gang's hideout. One of the Kids, Danny, returns to her stepfather's apartment to bring some clothes for her. He is arrested by the police, suspected of the murder. [ 'Neath Brooklyn Bridge ] [ / (Person of Interest) ] in <e:Champaign, Illinois>Champaign, Illinois</e>, Root (<e:Amy Acker>Amy Acker</e>) hijacks a prison bus and releases an inmate, Billy Parsons (<e:Colin Donnell>Colin Donnell</e>), per instruction from the Machine. She then gives him a fake ID so he infiltrates a government building posing as a <e:United States Department of Defense>United States Department of Defense</e> and receives a note with an address. She then leaves Billy to be arrested by the police. Radio <e:Stolytsia>Stolytsia</e> reported that Berkut riot police stopped a motorcade of protesters from heading towards the <e:Mezhyhirya Residence>presidential mansion in Mezhyhirya</e>, a suburb north of Kyiv. [ 1 December 2013 Euromaidan riots ] [ 1 July police stabbing ] [ 12 June 2019 Hong Kong protest ] [ 10 Rillington Place ] [ 1 July police stabbing ] [ 1 July police stabbing ] GEN Anna Joshi M BJP Pune GEN Anna Joshi M BJP 10th Lok Sabha 13.849536 Lok Sabha Pune [ GEN ]
>
> **MoRe prediction:** **Mumbai[LOC] &#x2716;** **BJP[ORG] &#x2716;** cadres stopped by Police from marching towards **Mantralaya[LOC] &#x2714;**. Arrested. #SackShinde
>
> **Gold label:** **Mumbai BJP[ORG]** cadres stopped by Police from marching towards **Mantralaya[LOC]**. Arrested. #SackShinde
> _______________________________________________________________________________________________
>
> You read that right, this is not garbled characters, this is indeed the data we sampled from the MoRe(Text) dataset. And the following is the additional information retrieved by PGIM from the GPT:
>
> _______________________________________________________________________________________________
> **Additional information from GPT:** Named entities:1. Sootradhar (person/handle) 2. Mumbai (location)  3. BJP (political party) 4. Police (law enforcement) 5. Mantralaya (government building) 6. #SackShinde (hashtag/issue)
>
> Reasoning: The sentence mentions a person or handle named Sootradhar who is likely tweeting about a political protest in Mumbai. **The Mumbai BJP refers to the local branch of the Bharatiya Janata Party, a major political party in India.** The police are mentioned as stopping the protesters from marching towards **Mantralaya, a government building in Mumbai.** The hashtag #SackShinde is likely a reference to a political issue or controversy, but without further context it is unclear what this refers to. The image of a crowd of people standing around a bus does not appear to be directly related to the sentence, but may be a part of the larger context of the political protest.
>
> **PGIM prediction:** **Mumbai BJP[ORG] &#x2714;** cadres stopped by Police from marching towards **Mantralaya[LOC] &#x2714;**. Arrested. #SackShinde
>
> **Gold label:** **Mumbai BJP[ORG]** cadres stopped by Police from marching towards **Mantralaya[LOC]**. Arrested. #SackShinde
> _______________________________________________________________________________________________
>
> Obviously, the reason why PGIM can make a correct prediction is that the information retrieved by GPT clearly states that "Mumbai BJP refers to the Bharatiya Janata Party". However, the information retrieved by MoRe(Text) from Wiki provides almost no assistance in the recognition of named entities.
>
> Then there is a case where MoRe(Text) made the right prediction but PGIM got it wrong:
>
> _______________________________________________________________________________________________
> **Additional information from wiki:** [Jaber Al-Ahmad Al-Sabah] <e:Sheikh>Sheikh</e> Jaber al-Ahmad al-Sabah (29 June 1926 – 15 January 2006) (<e:Arabic language>Arabic</e>: الشيخ جابر الأحمد الجابر الصباح) of the <e:Al-Sabah>al-Sabah</e> dynasty was the Emir of Kuwait and <e:Commander-in-chief>Commander</e> of the <e:Military of Kuwait>military of Kuwait</e> who served in the posts from 31 December 1977 until his death on 15 January 2006 due to <e:Cerebral hemorrhage>cerebral hemorrhage</e>. The third monarch to rule Kuwait since its <e:Independence>independence</e> from <e:British Empire>Britain</e>, Jaber had previously served as <e:Ministry of Finance (Kuwait)>minister of finance and economy</e> from 1962 to 1965 when he was appointed <e:Prime minister>prime minister</e> prior to becoming Kuwait's ruler. [newline_symbol] Main article: <e:Gulf War>Gulf War</e> [Nuclear Security Summit] The Nuclear Security Summit (NSS) is a world summit, aimed at preventing <e:Nuclear terrorism>nuclear terrorism</e> around the globe. The first summit was held in <e:Washington, D.C.>Washington, D.C.</e>, United States, on April 12–13, 2010. The second summit was held in <e:Seoul>Seoul</e>, South Korea, in 2012. The third summit was held in <e:The Hague>The Hague</e>, Netherlands, on March 24–25, 2014. The fourth summit was held in Washington, D.C. on March 31–April 1, 2016 [newline_symbol] Main article: <e:2010 Nuclear Security Summit>2010 Nuclear Security Summit</e>
>
> **MoRe prediction:** **Bush[PER] &#x2714;** 41, 43 Do Not Plan to Endorse **Donald Trump[PER] &#x2714;**.
>
> **Gold label:** **Bush[PER]** 41, 43 Do Not Plan to Endorse **Donald Trump[PER]**.
> _______________________________________________________________________________________________
> **Additional information from PGIM:** Named entities: 1. Bush 41 (person/former president) 2. Bush 43 (person/former president) 3. Donald Trump (person/presidential candidate).
>
> Reasoning: **The sentence mentions Bush 41 and 43, who are both former presidents of the United States.** The sentence also mentions Donald Trump, who is a current presidential candidate. The image of the word "declassify" does not seem to be related to the sentence, as it is not mentioned in the text or connected to any of the named entities.
>
> **PGIM prediction:** **Bush 41[PER] &#x2716;** , 43 Do Not Plan to Endorse **Donald Trump[PER] &#x2714;**.
>
> **Gold label:** **Bush[PER]** 41, 43 Do Not Plan to Endorse **Donald Trump[PER]**.
> _______________________________________________________________________________________________
> In this case, Bush is not a named entity that is hard to predict correctly. But since the additional knowledge retrieved by GPT clearly pointed out that Bush 41 is a name of person, PGIM ended up making a prediction that didn't match the gold label. This shows that the knowledge retrieved by PGIM from GPT may mislead the final prediction of the model in a few cases. On the one hand, even with numerous interventions, it is still difficult to make GPT always generate answers that conform to the annotation style of the dataset; on the other hand, in rare cases, GPT may generate wrong answers.
>
> But the reason why MoRe(Text) can make correct predictions is obviously not related to the information it retrieves from the wiki, because "Bush" is not even mentioned in its additional knowledge. In fact, after masking the noise retrieved from the wiki, RoBERTa+CRF can more easily predict correctly.
>
> In summary, considering the relevance and length of retrieved information, GPT has advantages over wiki in almost any case. The information retrieved from GPT is usually unambiguous and directional, which leads to it providing obvious help for prediction in most cases, but also potentially misleading prediction in rare cases.
>
> For reasons of space, we do not list further cases. All of the above analyzes will eventually be condensed and added to the paper. And we will select more similar cases and add them to the end of the Appendix of paper to help readers have a more intuitive feeling for the advantages of PGIM.
>
> Finally, we have fixed all the typos grammar style and presentation improvements. Thank you for your careful reading of our article. Your comments are very helpful for the improvement of our paper!
>
> Reference:
>
> [1] ITA: Image-Text Alignments for Multi-Modal Named Entity Recognition (NAACL 2022)
>
> [2] PromptMNER: Prompt-Based Entity-Related Visual Clue Extraction and Integration for Multimodal Named Entity Recognition (DASFAA 2022)
>
> [3] CAT-MNER: Multimodal Named Entity Recognition with Knowledge-Refined Cross-Modal Attention (ICME 2022)
>
> [4] Named Entity and Relation Extraction with Multi-Modal Retrieval (EMNLP 2022)
>
> [5] Improving multimodal named entity recognition via entity span detection with unified multimodal transformer (ACL 2020)
>
> [6] Multi-modal graph fusion for named entity recognition with targeted visual guidance (AAAI 2021)
>
> [7] MNER-QG: An End-to-End MRC Framework for Multimodal Named Entity Recognition with Query Grounding (AAAI 2023)
>
> [8] Minigpt-4: Enhancing vision-language understanding with advanced large language models （arXiv:2304.10592）
>
> [9] mPLUG-Owl: Modularization Empowers Large Language Models with Multimodality （arXiv:2304.14178）

---

### Official Review · Reviewer_eSeH · 2023-08-04

**Typos Grammar Style And Presentation Improvements:** N/A
**Soundness:** 3

**Excitement:**

3: Ambivalent: It has merits (e.g., it reports state-of-the-art results, the idea is nice), but there are key weaknesses (e.g., it describes incremental work), and it can significantly benefit from another round of revision. However, I won't object to accepting it if my co-reviewers champion it.

**Missing References:**

N/A

**Paper Topic And Main Contributions:**

This paper presents a two-stage framework that utilizes ChatGPT to generate auxiliary knowledge, enhancing the accuracy of Multimodal Named Entity Recognition (MNER). The paper conducts sufficient experiments, but some essential elements, such as the annotation process and examples of predefined artificial samples, lack detailed explanations. Additionally, the MSEA module proposed in the paper seems to have minimal impact on model performance.

**Questions For The Authors:**

N/A

**Reasons To Accept:**

1. The paper proposes a novel method for MNER and provides comprehensive experimental validation of PGIM. Leveraging ChatGPT for auxiliary knowledge generation results in improved generalization and robustness.
2. The experiments were conducted in a fairly comprehensive manner.
3. The motivation behind this approach, aiming to enhance the performance of MNER using external knowledge, is intuitive and well-founded.


**Reasons To Reject:**

1. The description of the notation process of predefined artificial samples is rather brief and lacks clarity. Such as,
(1) How many predefined artificial samples were used?
(2) How did the authors annotate samples in the dataset where image and text exhibit low correlation?

2. The findings presented in Table 3 suggest that the MSEA module has minimal impact on the model's performance, raising doubts about its utility. The observed performance improvement in the model can likely be attributed to the utilization of predefined artificial samples and a well-designed prompt.

3. The author mentions the Baseline in line 526, but there is no detailed explanation of its structure. Although lacking auxiliary knowledge, the Baseline still demonstrates commendable performance.


**Reproducibility:**

3: Could reproduce the results with some difficulty. The settings of parameters are underspecified or subjectively determined; the training/evaluation data are not widely available.

**Reviewer Confidence:**

3: Pretty sure, but there's a chance I missed something. Although I have a good feel for this area in general, I did not carefully check the paper's details, e.g., the math, experimental design, or novelty.

---

> ### Author Rebuttal · Authors · 2023-08-28
>
> Thank you for your thoughtful comments on our work. And thank you for your affirmation of the research motivation and comprehensiveness of the experiment. In the following, we reply to specific points raised by the reviewer.
>
> **Q1: The description of the notation process of predefined artificial samples is rather brief and lacks clarity. Such as, (1) How many predefined artificial samples were used? (2) How did the authors annotate samples in the dataset where image and text exhibit low correlation?**
>
> A1: First, about predefined artificial samples: specifically, for Twitter-2017 dataset, we randomly sample 200 samples from the training set for manual labeling, and for Twitter-2015 dataset, the number is 120. The annotators consist of three NLP graduate students with backgrounds in NER and LLMs. For many possibilities encountered in the labeling process, what the annotator needs to do is to correctly judge and interpret the sample from the perspective of humans. For samples where images and text are related, we directly state which entities in the text are emphasized by the image. For samples where the image and text are unrelated, we directly declare that the image description is unrelated to the text. We select a few examples as follows:
> _______________________________________________________________________________________________
> Sample ID: 16_05_24_110
>
> Named entities: 1. The Cure (band)
> Reasoning: The sentence mentions "The Cure" which is a famous rock band. **The image may not be related to the band the sentence mentioned.**
>
> Sample ID: O_1313
>
> Named entities: 1. Justin Bieber (person/celebrity) 2. New York City (place/city)
> Reasoning: The sentence mentions Justin Bieber, a well-known celebrity and musician. New York City is a well-known city in the United States and is mentioned because that is where Bieber was spotted. **The picture has nothing to do with the sentence.**
> _______________________________________________________________________________________________
> Additionally, after the anonymity period is over, we will upload all paper-related data (including code, the dataset constructed using GPT, and predefined artificial samples) to GitHub for open access. Currently, you can view it through Supplementary Materials.
>
> Your valuable comments made us realize that a more detailed explanation of the notation process of predefined artificial samples can better help readers understand paper. So, the above replies will eventually be condensed in the camera-ready version and added to paragraph 3.2 Predefined artificial samples.
>
> **Q2: The findings presented in Table 3 suggest that the MSEA module has minimal impact on the model's performance, raising doubts about its utility. The observed performance improvement in the model can likely be attributed to the utilization of predefined artificial samples and a well-designed prompt.**
>
> A2: First, the effectiveness of the MSEA module is positively correlated with the quantity and quality of predefined artificial samples. The presence or absence of the MSEA module will lead to the utilization of distinct predefined samples for constructing prompts for GPT. When randomly selecting samples to feed into GPT, the significance of these samples primarily lies in guiding GPT to generate responses with a controllable format. The difference from the former is that the most similar samples selected by the MSEA module are more likely to assist GPT not only in attending to the response format but also in roughly predicting the correct entities. This is because the answer is highly likely to be embedded within the predefined samples retrieved by MSEA. Evidently, the greater the number of predefined artificial samples, the more likely the MSEA module is to create a scenario where the answer is hidden within the prompt.
>
> In summary, we reasonably believe that increasing the number of manually labeled samples will make the MSEA module more effective. However, since the original intention of PGIM is to allow the ability of LLM to be used reasonably in MNER and to explore the possibility of obtaining marginal benefits in performance through the MSEA module + a small number of predefined artificial samples. Therefore, after the experiments in Table 3 are sufficient to prove that the introduction of MSEA not only does not damage the accuracy, but also brings a certain degree of accuracy improvement, we did not further invest too many artificial samples. Because we think that the small improvement in accuracy is far less important than the research motivation we want to convey. However, how to make the prompt module such as MSEA produce greater effect with less cost is a new problem worth studying in the future.
>
> Finally, it is worth noting that whether it is the predefined artificial samples, well-designed MNER prompts, or the design of the MSEA module, all are part of the contribution of this paper. The importance of the three parts may be different, but the combination of the three is the final contribution that this paper wants to convey.
>
> **Q3: The author mentions the Baseline in line 526, but there is no detailed explanation of its structure. Although lacking auxiliary knowledge, the Baseline still demonstrates commendable performance.**
>
> A3: This Baseline refers to: compared with the complete PGIM, auxiliary knowledge is not added in any way, and only the original sentence is retained. Therefore, in order to control variables, the other settings of Baseline are exactly the same as PGIM, including using XLM-RoBERTa-large as the text encoder. And as shown in the "Text" section in Table 1, replacing BERT with RoBERTa can achieve a 3% performance improvement. This may be an important reason for your mention of "commendable baseline performance".
>
> The reason why the reviewers have this doubt may be that the expression in line 526 of the paper is not clear enough. We will simply modify line 526 of the paper to make the expression clearer.

---

### Official Review · Reviewer_KB7s · 2023-08-04

**Soundness:** 3

**Excitement:**

3: Ambivalent: It has merits (e.g., it reports state-of-the-art results, the idea is nice), but there are key weaknesses (e.g., it describes incremental work), and it can significantly benefit from another round of revision. However, I won't object to accepting it if my co-reviewers champion it.

**Paper Topic And Main Contributions:**

In this paper, the author argued that existing studies mainly focus on maximizing the utilization of pertinent image information or incorporating external knowledge from explicit knowledge bases. However, these methods still suffer from two problems: (1) neglect the necessity of providing the model with external knowledge; (2) high redundancy in the retrieved knowledge. To tackle the above issues, they proposed a two-stage framework PGIM that aims to leverage ChatGPT to heuristically generate auxiliary knowledge for more efficient entity prediction. Experiments on two MNER datasets show their model achieves state-of-the-art performance. Further analysis verifies the effectiveness of the method.

pros:

(1) Well-organized and well-written.

(2) New framework.

(3) sufficient experimental analysis.

cons:

(1) Unfair comparison.

(2) marginal improvement compared with MoRe and the standard deviation is not provided.

(3) some results have bias.


**Reasons To Accept:**

(1) The paper is well-organized and well-written.

(2) They design a new framework to deal with the MNER task, i.e., leverage ChatGPT to generate auxiliary knowledge for more efficient entity prediction.

(3) The experimental analysis and comparison are comprehensive, and the results are competitive.


**Reasons To Reject:**

(1) As the author mentioned in Model Configuration, they use the XLM-RoBERTa_large as encoder, but as far as I know, most of the previous methods used Bert-base as the encoder, so whether it is a fair comparison is also a concern. I suggest adding experiments with Bert-base as the encoder.

(2) As shown in Table 1, the improvement of some results over the compared baseline MoRe is not so great, so I would like to see the standard deviation of these results. Besides, no significance tests are conducted.

(3) As shown in Table 3, the authors explore the effect of the number of in-context examples on the test set rather than on the dev set, which brings bias to the proposed model.


**Reproducibility:**

4: Could mostly reproduce the results, but there may be some variation because of sample variance or minor variations in their interpretation of the protocol or method.

**Reviewer Confidence:**

4: Quite sure. I tried to check the important points carefully. It's unlikely, though conceivable, that I missed something that should affect my ratings.

---

> ### Author Rebuttal · Authors · 2023-08-28
>
> Thank you for your thoughtful comments on our work. And thank you for your affirmation of the pros of our paper. We will address your concerns as follows:
>
> **Q1: As the author mentioned in Model Configuration, they use the XLM-RoBERTa-large as encoder, but as far as I know, most of the previous methods used Bert-base as the encoder, so whether it is a fair comparison is also a concern. I suggest adding experiments with Bert-base as the encoder.**
>
> A1: First, we would like to emphasize the fairness of experimental comparison, the detailed reasons are as follows:
>
> The experimental results of all the competitive MNER methods (e.g., ITA[1]，PromptMNER[2]，CAT-MNER[3]，MoRe[4]) shown in Table 1 in the past two years, their text encoders are all RoBERTa instead of BERT. In fact, ITA is a dividing line. Starting from the ITA[1] in 2022, MNER research has all chosen RoBERTa as the encoder. It is precisely because of this that, as shown in Table 1, after ITA[1], the accuracy of the MNER methods has achieved a leapfrog performance improvement compared with the "previous methods used Bert" you mentioned (e.g., UMT[5], UMGF[6], R-GCN[7]).
>
> As shown in the "Text" part of Table 1, replacing BERT with RoBERTa can achieve a performance improvement of about 3%. Therefore, in order to make a fair comparison, PGIM can only follow the settings of these advanced MNER methods (ITA[1]，PromptMNER[2]，CAT-MNER[3]，MoRe[4]) in the past and use RoBERTa. It is also under this fair experiment with a fixed encoder that PGIM shows better performance than other methods, which reflects the innovation and effectiveness of our method.
>
> However, thank you very much for your thoughtful suggestion! We had even more surprising findings in supplementary experiments. We performed experiments by replacing the encoders of all open-source RoBERTa-large MNER methods with BERT-base. Text encoders for all methods in Table 6 are BERT-base. Baseline$_{BERT}$ represents inputting original samples into BERT+CRF. All of the results are averaged from 3 runs with different random seeds. The marker * refers to significant test p-value < 0.05 when comparing with CAT-MNER and MoRe(Image/Text). The experimental results show that PGIM not only continues SOTA performance, but also achieves a greater performance improvement than the RoBERTa version experiment, especially on the Twitter-2017 dataset.
>
> |           Table 6     |  |**Twitter-2015**     |     | |    **Twitter-2017**  |     |
> |-----------------------|:----------------:|:---:|:---:|:----------------:|:---:|:---:|
> |                       |       Pre.       | Rec.|  F1  |       Pre.       | Rec.|  F1  |
> | Baseline$_{BERT}$ |      75.56      |73.88|74.71 |      86.10       |83.85|84.96 |
> | UMT                   |      71.67      |75.23|73.41 |      85.28       |85.34|85.31 |
> | UMGF                  |      74.49      |75.21|74.85 |      86.54       |84.50|85.51 |
> | R-GCN                 |      73.95      |76.18|75.00 |      86.72       |87.53|87.11 |
> | ITA$_{BERT}$   |        -        |  -  |75.60 |        -         |  -  |85.72 |
> | CAT-MNER$_{BERT}$ |    **76.19**    |74.65|75.41 |      87.04       |84.97|85.99 |
> | MoRe$_{Image BERT}$ |  73.16  |74.64|73.89 |      85.49       |86.38|85.94 |
> | MoRe$_{Text BERT}$ |  73.31  |74.43|73.86 |      85.92       |86.75|86.34 |
> | PGIM$_{BERT}$  |      75.84      |**77.76**|**76.79*** |     **89.09**      |**90.08**|**89.58*** |
> |                       | $\pm$0.30 |$\pm$0.22|$\pm$0.19 | $\pm$0.24 |$\pm$0.08|$\pm$0.10|
>
> We think the reason for this phenomenon is as follows: RoBERTa conceals the defects of previous MNER methods through its strong encoding ability, and these defects are further amplified after using BERT. For example, the encoding ability of BERT on long text is weaker than RoBERTa, and the additional knowledge retrieved by MoRe (Image/Text) is much longer than PGIM. Therefore, as shown in Table 4 and Table 6, the performance loss of MoRe (Image/Text) is larger than the performance loss of PGIM after replacing BERT.
>
> We will add this experiment and analysis to the experimental section of paper. In addition, in order to avoid similar doubts for readers, we will add a supplementary statement in the 4.1 Model Configuration chapter -- "For a fair comparison, PGIM chooses to use the same text encoder XLM-RoBERTa-large as ITA[1], PromptMNER[2], CAT-MNER[3] and MoRe[4]."
>
> **Q2: As shown in Table 1, the improvement of some results over the compared baseline MoRe is not so great, so I would like to see the standard deviation of these results. Besides, no significance tests are conducted.**
>
> A2: First, your insights on standard deviation and significance testing are quite valuable. According to statistics, the standard deviation of PGIM in Table 1 are as follows. And the marker * refers to significant test p-value < 0.05 when comparing with CAT-MNER and MoRe(MoE):
>
> | Tabel 1${_{add SD}}$ | ******|Twitter|-   2015 |******|******|******|****** | ******|Twitter|- 2017  |****** |****** |****** | ******|
> | --- | --- | --- | --- | --- | --- | --- | --- | --- | --- | --- | --- | --- | --- | --- |
> |  | | Single|  Type(F1)| | |Overall | | |Single |Type(F1) | | Overall | | |
> | | PER | LOC  | ORG | OTH. | Pre. | Rec. | F1 | PER | LOC | ORG | OTH. | Pre. | Rec. | F1 |
> | | | | | | | |Text+Image | | | | | | | |
> | UMT | 85.24    | 81.58 | 63.03 | 39.45 | 71.67 | 75.23 | 73.41 | 91.56 | 84.73 | 82.24 | 70.10 | 85.28 | 85.34 | 85.31 |
> | UMGF | 84.26 | 83.17 | 62.45 | 42.42 | 74.49 | 75.21 | 74.85 | 91.92 | 85.22 | 83.13 | 69.83 | 86.54 | 84.50 | 85.51 |
> | MNER-QG | 85.68 | 81.42 | 63.62 | 41.53 | 77.76 | 72.31 | 74.94 | 93.17 | 86.02 | 84.64 | 71.83 | 88.57 | 85.96 | 87.25 |
> | R-GCN | 86.36 | 82.08 | 60.78 | 41.56 | 73.95 | 76.18 | 75.00 | 92.86 | 86.10 | 84.05 | 72.38 | 86.72 | 87.53 | 87.11 |
> | ITA | - | - | - | - | - | - | 78.03 | - | - | - | - | - | - | 89.75 |
> | PromptMNER | - | - | - | - | 78.03 | 79.17 | 78.60 | - | - | - | - | 89.93 | 90.60 | 90.27 |
> | CAT-MNER | 88.04 | **84.70** | 68.04 | 52.33 | 78.75 | 78.69 | 78.72 | 94.61 | 88.40 | 88.14 | **80.50** | 90.27 | 90.67 | 90.47 |
> | MoRe${_{MoE}}$ | - | - | - | - | - | - | 79.21 | - | - | - | - | - | - | 90.67 |
> | PGIM(Ours) | **88.34** | 84.22 | **70.15** | **52.34** | **79.21** | **79.45** | **79.33*** | **96.46** | **89.89** | **89.03** | 79.62 | **90.86** | **92.01** | **91.43*** |
> | | $\pm$0.02 | $\pm$0.12 | $\pm$0.36 | $\pm$0.98 | $\pm$0.63 | $\pm$0.22 | $\pm$0.06 | $\pm$0.02 | $\pm$0.68 | $\pm$0.53 | $\pm$2.25 | $\pm$0.16 | $\pm$0.07 | $\pm$0.09 |
>
> In fact, we have indeed attempted to display the standard deviation in the Table 1, as it holds significant importance. But considering that most of the methods in Table 1 (UMT[5], UMGF[6], MNER-QG[8], ITA[1], PromptMNER[2], CAT-MNER[3], MoRe[4]) did not report the standard deviation in their paper, we ended up following this "convention" in MNER.
>
> This problem exists in almost all the work in MNER field. But we hope that PGIM can be the terminator of this problem, as a solid baseline in the field of MNER. So, we will modify Table 1 in the camera-ready version to clearly present the standard deviation and significance test results.
>
> Second, we need to point out that the fair opponent of PGIM should be MoRe (Image/Text) in the Appendix Table 4, because they have the same macrostructure, same parameter quantity and same encoder (RoBERTa-large). The only difference is the method of obtaining additional information and the quality of the acquired knowledge.
>
> What Table 1 really wants to reflect is: as we stated in lines 833-846 of the paper, even when making an unfair comparison with MoRe (MoE), which is built upon MoRe (Image+Text) by incorporating post-processing steps-MoE (This means that MoRe[4] needs to train three models: MoRe(Image)+MoRe(Text)+MoRe(MoE)). PGIM still achieves better results without any excessive complex design (Just need to train one model).
>
> In other words, compared with MoRe (Image/Text) with the same architecture and parameter quantity, PGIM has significant performance advantages (as shown in Table 4). Compared with the complete MoRe (MoE), PGIM still achieves better results with a simpler and lighter model.
>
> Your suggestion also made us realize that when we compare PGIM with MoRe(Image/Text) or MoRe(MoE), we should explain the relationship between them more clearly. Above analyzes will eventually be condensed and added to the Appendix of the paper in camera-ready version.
>
> **Q3: As shown in Table 3, the authors explore the effect of the number of in-context examples on the test set rather than on the dev set, which brings bias to the proposed model.**
>
> A3: First, throughout this paper, the metrics reported by PGIM are the results of the test set. For all methods (UMT[5], UMGF[6], R-GCN[7], MNER-QG[8], ITA[1], PromptMNER[2], CAT-MNER[3], MoRe[4]) compared with PGIM, their reports are also test set results. And most researchers agree that reporting test set accuracy is considered convincing in most deep learning tasks. Given the writing conventions in the field, we did not report the metrics on the dev set in paper.
>
> However, out of respect to the reviewer, we recounted the effect of the number of in-context examples on the dev set. Since only the hyperparameters yielding the highest accuracy on the dev set are employed for a single test on the test set and given that the disparity between the dev and test data in both datasets used in our experiments is not substantial, the secondary results do not exhibit a significant distinction. The observed phenomena in the results still adhere to the viewpoints we conveyed in the original text, spanning lines 521 to 548.
>
>
> |         Tabel 7          | Twitter-2015 |       |       | Twitter-2017 |       |       |
> |-------------------|--------------|-------|-------|--------------|-------|-------|
> |                   | Pre.         | Rec.  | F1    | Pre.         | Rec.  | F1    |
> | Baseline          | 76.25        | 78.22 | 77.22 | 88.75        | 90.56 | 89.64 |
> | w/o MSEA$_{N=1}$  | 77.90        | 78.83 | 78.36 | 90.44        | 90.79 | 90.61 |
> | w/o MSEA$_{N=5}$  | 78.34        | 79.32 | 78.83 | 90.54        | 91.77 | 91.15 |
> | w/o MSEA$_{N=10}$ | 77.47        | 79.90 | 78.66 | 90.59        | 91.93 | 91.26 |
> | PGIM$_{N=1}$      | 77.44        | 80.18 | 78.79 | 90.34        | 91.41 | 90.88 |
> | **PGIM$_{N=5}$**  | 78.41        | **80.53** | **79.45** | 90.67        | **92.82** | **91.73** |
> | PGIM$_{N=10}$     | **78.69**    | 79.77 | 79.23 | **90.97**    | 92.07 | 91.52 |
>
>
> The above constitutes our complete response to the concerns you raised, with the hope of resolving any uncertainties you had. We would greatly appreciate it if you could further engage in a more comprehensive objective assessment of the contributions and the soundness of this paper. Because as far as we know, PGIM represents the pioneering attempt in the era of LLMs to incorporate LLM for addressing MNER challenges. We contend that this work possesses considerable inspiration for the pertinent field and it deserves to be discovered by more NER researchers.
>
> **Reference:**
>
> [1] ITA: Image-Text Alignments for Multi-Modal Named Entity Recognition (NAACL 2022)
>
> [2] PromptMNER: Prompt-Based Entity-Related Visual Clue Extraction and Integration for Multimodal Named Entity Recognition (DASFAA 2022)
>
> [3] CAT-MNER: Multimodal Named Entity Recognition with Knowledge-Refined Cross-Modal Attention (ICME 2022)
>
> [4] Named Entity and Relation Extraction with Multi-Modal Retrieval (EMNLP 2022)
>
> [5] Improving multimodal named entity recognition via entity span detection with unified multimodal transformer (ACL 2020)
>
> [6] Multi-modal graph fusion for named entity recognition with targeted visual guidance (AAAI 2021)
>
> [7] Learning from Different text-image Pairs: A Relation-enhanced Graph Convolutional Network for Multimodal NER (ACMMM 2022)
>
> [8] MNER-QG: An End-to-End MRC Framework for Multimodal Named Entity Recognition with Query Grounding (AAAI 2023)

---

### Meta-Review · Area_Chair_LUAC · 2023-09-21

**Recommendation:** 4

**Metareview:**

This paper presents a two-stage framework PGIM that utilizes ChatGPT to generate auxiliary knowledge for Multimodal Named Entity Recognition (MNER). Experimental analysis and comparison are comprehensive. Important details, such as the annotation process and examples, are missing in the submitted version (provided in the rebuttal). The performance gains of the MSEA module are somehow slight.

---

### Decision · Program_Chairs · 2023-10-07

**Decision:**

Accept-Findings

**Comment:**

This paper presents a two-stage framework PGIM that utilizes ChatGPT to generate auxiliary knowledge for Multimodal Named Entity Recognition (MNER). Experimental analysis and comparison are comprehensive. Important details, such as the annotation process and examples, are missing in the submitted version (provided in the rebuttal). The performance gains of the MSEA module are somehow slight.